# The cell-wide web coordinates cellular processes by directing site-specific Ca$^{2+}$ flux across cytoplasmic nanocourses

Jingxian Duan[1], Jorge Navarro-Dorado[1], Jill H. Clark[1], Nicholas P. Kinnear[1], Peter Meinke[2], Eric C. Schirmer [2] & A. Mark Evans[1]

Ca$^{2+}$ coordinates diverse cellular processes, yet how function-specific signals arise is enigmatic. We describe a cell-wide network of distinct cytoplasmic nanocourses with the nucleus at its centre, demarcated by sarcoplasmic reticulum (SR) junctions ($\leq$400 nm across) that restrict Ca$^{2+}$ diffusion and by nanocourse-specific Ca$^{2+}$-pumps that facilitate signal segregation. Ryanodine receptor subtype 1 (RyR1) supports relaxation of arterial myocytes by unloading Ca$^{2+}$ into peripheral nanocourses delimited by plasmalemma-SR junctions, fed by sarco/endoplasmic reticulum Ca$^{2+}$ ATPase 2b (SERCA2b). Conversely, stimulus-specified increases in Ca$^{2+}$ flux through RyR2/3 clusters selects for rapid propagation of Ca$^{2+}$ signals throughout deeper extraperinuclear nanocourses and thus myocyte contraction. Nuclear envelope invaginations incorporating SERCA1 in their outer nuclear membranes demarcate further diverse networks of cytoplasmic nanocourses that receive Ca$^{2+}$ signals through discrete RyR1 clusters, impacting gene expression through epigenetic marks segregated by their associated invaginations. Critically, this circuit is not hardwired and remodels for different outputs during cell proliferation.

[1] Centres for Discovery Brain Sciences and Cardiovascular Sciences, College of Medicine and Veterinary Medicine, Hugh Robson Building, University of Edinburgh, Edinburgh EH8 9XD, UK. [2] Wellcome Centre for Cell Biology, Michael Swann Building, University of Edinburgh, Edinburgh EH9 3BF, UK. Correspondence and requests for materials should be addressed to A.M.E. (email: mark.evans@ed.ac.uk)

Cells select for one or a combination of distinct functions through Ca$^{2+}$ signalling[1]. Therefore, stimuli must induce different Ca$^{2+}$ signals to engage specific cellular responses, such as, for example, contraction or relaxation of smooth muscles as well as their switch from contractile to migratory-proliferative phenotypes, which additionally requires changes in gene expression[2]. However, despite the extraordinarily detailed mapping of the temporal characteristics of both unitary and macroscopic Ca$^{2+}$ signals across a variety of cell types[3–8], how cells deliver the diverse range of site- and function-specific Ca$^{2+}$ signals necessary to coordinate the full panoply of cellular processes remains enigmatic[2].

The primary intracellular Ca$^{2+}$ store is the sarco/endoplasmic reticulum (S/ER)[2], which is known to be a contiguous organelle, from its origin at the outer nuclear membrane (ONM) to the periphery of the cell. Yet the S/ER delivers Ca$^{2+}$ signals with clear diversities of form and function[4,5]. In arterial smooth muscles, for example, the current consensus is that relaxation is mediated by highly localised Ca$^{2+}$ sparks that recruit Ca$^{2+}$-activated potassium channels to promote plasma membrane hyperpolarization, while contraction is triggered by propagating global Ca$^{2+}$ waves[9], with adjustments to gene expression presumed to be governed by the spatiotemporal patterns of global Ca$^{2+}$ transients that gain unrestricted entry to the nucleoplasm across the nuclear envelope (NE) and its invaginations[10–14].

However, in smooth muscles it has long been suggested that multiple, spatially segregated and independently releasable sub-compartments of Ca$^{2+}$ may exist within the SR, filled by spatially segregated subtypes of sarco/endoplasmic reticulum Ca$^{2+}$ ATPase (SERCA) pumps and mobilised through similarly segregated subtypes of Ca$^{2+}$ release channel[15–17], including ryanodine receptors (RyRs) 1, 2 and 3[18–22]. This led to an alternative proposal, that different Ca$^{2+}$ signals may arise in distinct cytoplasmic spaces demarcated by junctions between the SR and its target organelles[23,24]. Hitherto, direct visualisation of Ca$^{2+}$ signalling within junctional complexes of the SR has not been achieved, so little more than speculation has guided such considerations on functional signal segregation within cells[23].

Here we identify a cell-wide network of distinct cytoplasmic nanocourses, demarcated both by the SR and by different types of SR resident Ca$^{2+}$ transporters and release channels. This extends from the plasma membrane to NE invaginations, delivering highly localised Ca$^{2+}$ flux, with path lengths on the nanoscale at all points. Cells may thus support unforeseen levels of network activity.

## Results

**Region-specific targeting of SERCA and RyRs**. We and others have previously demonstrated that within pulmonary arterial myocytes the peripheral SR proximal to the plasma membrane preferentially incorporates dense clusters of ryanodine receptor subtype 1 (RyR1) and of S/ER Ca$^{2+}$ ATPase subtype 2b (SER-CA2b), while RyR2s are incorporated in the extraperinuclear SR, where SERCA are few, with SERCA2a and RyR3 clusters heavily restricted to the deepest perinuclear SR[17,19]. Adding to this, we now identify within the same image series a distinct subtype of SERCA pump, SERCA1 (Fig. 1a, c), which together with a discrete subset of RyR1s (Fig. 1b, c) appear to line tubular networks within the boundary of the nucleus (Fig. 1a, b; Supplementary Fig. 1). This suggested that SERCA1s and RyR1s might line invaginations of the NE, the lumen of which is continuous with the SR.

**SR junctions demarcate networks of cytoplasmic nanocourses**. We explored this possibility using Fluo-4 to report on Ca$^{2+}$ signals and a membrane permeant DNA marker (Draq5) to identify the boundary of the nucleus. By adjusting the threshold for fluorescence detection, it was evident that a network of narrow tributaries of cytoplasm ≤500 nm wide penetrated the nucleus, perhaps reflecting the path of nuclear invaginations. The cytoplasmic nanocourses within the boundary of the Draq5 labelled nucleus exhibited markedly higher levels of Fluo-4 fluorescence than the surrounding nucleoplasm, and could be equally well distinguished from any aspect of the wider cytoplasm, which in turn and invariably exhibited higher basal fluorescence than the nucleoplasm (Fig. 2a). This suggested that invaginations of the NE might demarcate discrete signalling compartments that could be observed without the need for further image processing, irrespective of whether or not differences in fluorescence intensity resulted from differences in local cytoplasmic Ca$^{2+}$ concentration or the influence of the local environment within each of these compartments on general Fluo-4 fluorescence characteristics[25,26]. However, with the confocal system (see Methods) set to detect these cytoplasmic nanocourses within the boundary of the nucleus, we frequently observed variegated, region-specific differences in Fluo-4 fluorescence intensity in the bulk cytoplasm (beyond the boundary of the nucleus), i.e. highly localised, time-dependent and asynchronous fluctuations in Fluo-4 fluorescence intensity were evident across the wider cell. Therefore, we carried out deconvolution of all Fluo-4 images within each time series acquired. This revealed a cell-wide network of well-defined cytoplasmic nanocourses (≤400 nm across; Fig. 2a, threshold and $F_{max}$ set to highlight nanocourses; Supplementary Movie 1) that appeared to be demarcated by the SR (Supplementary Fig. 2). In short, different cytoplasmic nanocourses exist proximal to the plasma membrane, within extraperinuclear and perinuclear regions of cells and also penetrate the nucleus (identified by Draq5, blue, Fig. 2a). During short time series' (2–6 min; note, experiment duration limited by photo-toxicity) hotspots of local Ca$^{2+}$ flux, ≈200–400 nm in diameter, were readily identifiable in pseudocolour representations of this cell-wide network at rest (Fig. 2b), the fluorescence signal intensity of which oscillated without propagating beyond the nanocourse within which they arose. Each individual hotspot of Ca$^{2+}$ flux exhibited asynchronous temporal characteristics when compared to adjacent hotspots within the same nanocourse, or hotspots arising in different nanocourses (Fig. 2b, c; Supplementary Movies 2–4). Such activity was not evident in averages of Fluo-4 fluorescence for any given nanocourse as a whole (Fig. 2c, lower panels). Notably, distances of separation between hotspots for subplasmalemmal (359 ± 15 nm) and nuclear nanocourses (350 ± 13 nm; Fig. 2d) are consistent with those for skeletal muscle RyR1s[27], while distances of separation for extraperinuclear (414 ± 22 nm) and perinuclear (452 ± 32 nm) nanocourses (Fig. 2d) are significantly greater and consistent with those for cardiomyocyte RyR2s (0.6–0.8 μm)[28]. This is in accordance with previous studies on the spatial organisation of RyRs in the cells studied here (Fig. 1b, c). Hotspots of fluorescence were markedly attenuated by prior depletion of SR Ca$^{2+}$ stores using thapsigargin, a SERCA inhibitor (Fig. 2e and Supplementary Fig. 3), and upon blocking RyRs with tetracaine (Fig. 2e; note, following pre-incubation with tetracaine hotspots within nuclear nanocourses remained visible, but neither hotspots nor nanocourses could be reliably identified outside the boundary of the nucleus, where measures had to be taken within regions of interest under these conditions). As indicated above, nanocourse hotspots exhibited clear, time-dependent fluctuations in fluorescence intensity (Fig. 2f and Supplementary Fig. 4). Irrespective of the nanocourse in question, at least two discrete levels of hotspot intensity ($\Delta F_H/F_{N0}$; H = hotspot; $N_0$ = nanocourse at 0 s) were evident despite the limited temporal resolution at the optimal

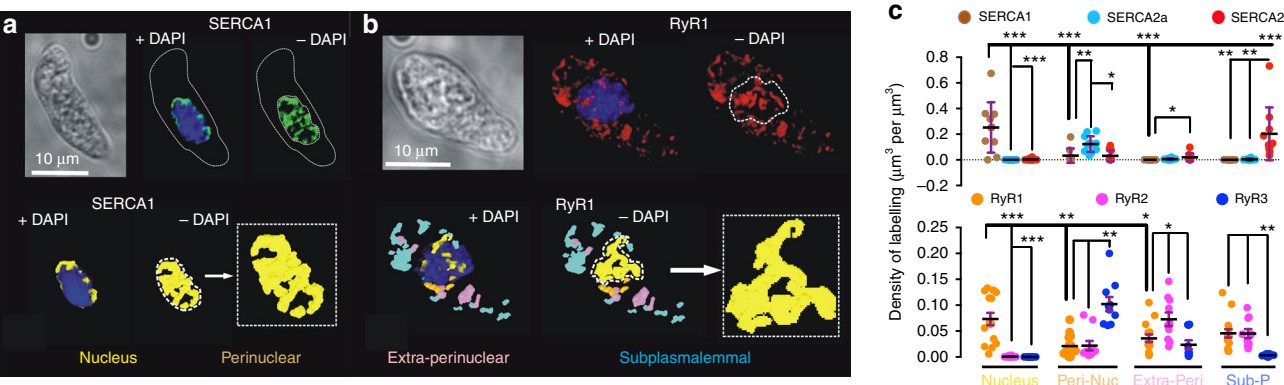

**Fig. 1** Discrete clusters of SERCA1 and RyR1 are targeted to the nuclear envelope. **a** Upper panels, left to right, bright field image of an arterial myocyte and 3D deconvolved fluorescence images (Deltavision, doconvolution) of SERCA1 labelling (green). Lower panels, left to right, digital skin encapsulating SERCA1 labelling ± nuclear labelling (blue, DAPI). **b** As for (**a**) but for RyR1 labelling (red). **c** Dot plot shows density of labelling ($\mu m^3$ per $\mu m^3$, mean ± SEM) for (upper panels) SERCA1 ($n = 10$ cells from 3 rats), SERCA2a ($n = 12$ cells from 3 rats) and SERCA2b ($n = 10$ cells from 3 rats) and (lower panels) RyR1 ($n = 15$ cells from 3 rats), RyR2 ($n = 12$ cells from 3 rats) and RyR3 ($n = 10$ cells from 3 rats) within the 4 designated regions of the cell; one-way ANOVA followed by a Tukey post-hoc test: *$p < 0.05$, **$p < 0.01$, ***$p < 0.001$

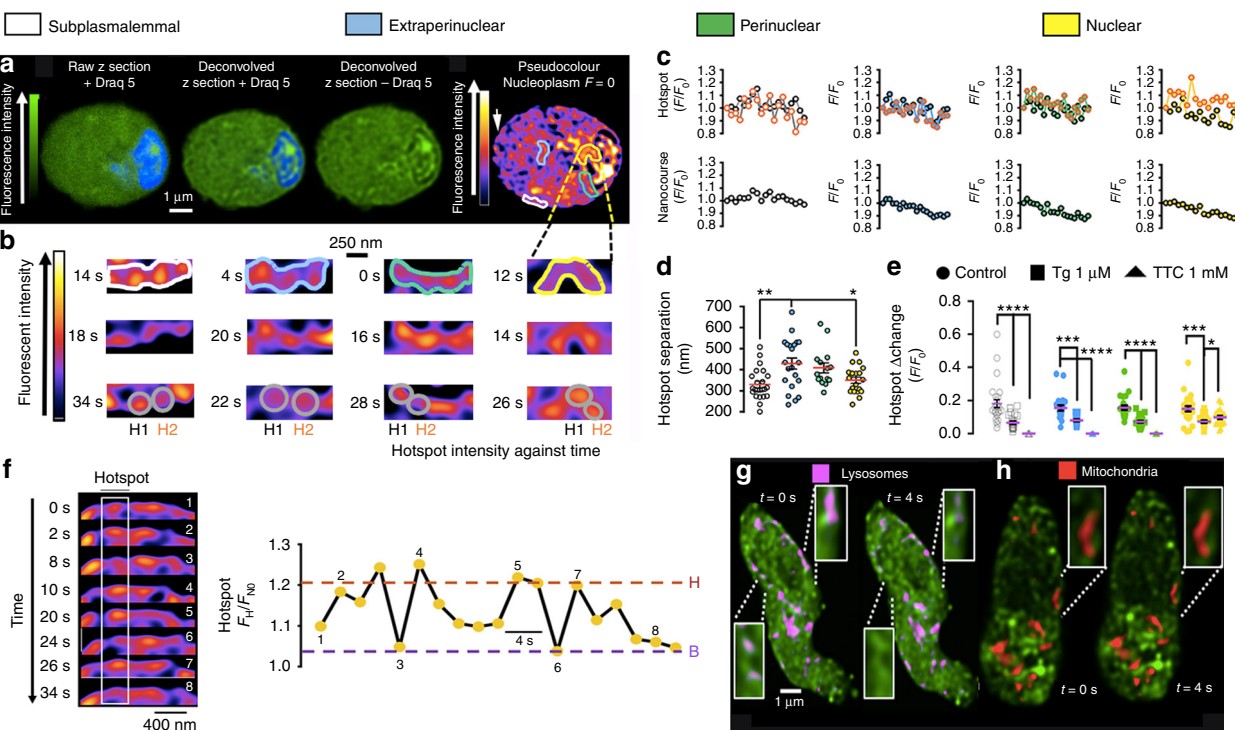

**Fig. 2** SR $Ca^{2+}$ flux within a cell-wide circuit of cytoplasmic nanocourses of arterial myocytes. **a** (left to right), confocal z sections through acutely isolated arterial myocyte loaded with Fluo-4 (green, calcium indicator) and Draq 5 (blue, nucleus), then deconvolved, and pseudocolor applied to show relative Fluo-4 intensity. Regions of interest identify exemplar subplaslemmal (white), extraperinuclear (blue), perinuclear (green) and nuclear (yellow) nanocourses. **b** (left to right), nanocourses in (**a**) at higher magnification and different time points; note, thresholds set independently to visualise hotspots. Grey circles identify hotspots (H1, black; H2, orange) of $Ca^{2+}$ flux in exemplar nanocourses. **c** Fluo-4 fluorescence ratio ($F_x/F_0$; where $F_0 =$ fluorescence at time 0 and $F_x =$ fluorescence at time = x) versus time (sampling frequency = 0.5 Hz) for H1 and H2 (upper panels, left to right) and the average for the whole nanocourse (lower panels, left to right). **d** Scatter plot shows distances separating hotspots (mean ± SEM; ≥36 hotspots, $n = 7$ cells from 7 rats) within subplasmalemmal (white), extraperinuclear (blue), perinuclear (green) and nuclear (yellow) nanocourses. **e** Dot plots show the effect of thapsigargin (1 μM; 30 min pre-incubation; $n = 3$ cells from 3 rats) and tetracaine (1 mM; 4 h pre-incubation; $n = 5$ cells from 4 rats) on the amplitude (mean ± SEM) of Fluo-4 fluorescence ratio change ($\Delta F_x/F_0$); t-test with Welch's correction: *$p < 0.05$, **$p < 0.01$, ***$p < 0.001$, ****$p < 0.0001$. **f** Image time series highlights (white rectangle) time-dependent intensity fluctuation of one hotspot in a different subplasmalemmal nanocourse (arrow in (**a**), upper panel, right most image), with a record of fluorescence intensity against time ($\Delta F_H/F_{N0}$; H = hotspot, $N_0$ = nanocourse at time = 0) from basal (B) to high intensity (H) states; note prolonged sub-state. **g** Deconvolved time series of z sections (0.25 Hz) show LysoTracker Red labelled endolysosomes in cytoplasmic nanocourses identified by Fluo-4 (confirmed in 3 cells from 3 different animals). **h** As for (**g**), but for mitochondria labelled with MitoTracker Red (confirmed in 3 cells from 3 different animals). Pseudocolour look up tables in (**a**) and (**b**) indicate relative fluorescence intensity in arbitrary units

sampling frequencies used here (0.5 Hz; scan speed limited by signal-to-noise). Transitions from basal to the highest level of fluorescence intensity ($\Delta F_x/F_0$, mean ± SEM: peripheral 0.18 ± 0.02; extraperinuclear 0.15 ± 0.01; perinuclear 0.16 ± 0.02; nuclear 0.15 ± 0.02; $n = 7$ cells from 7 rats) varied in duration from ~2 s to ≥10 s, with even longer dwell times evident for lower frequency, low intensity sub-states. The asynchronous activity, spatial characteristics and pharmacology of hotspots suggests that these events most likely reflect low level, basal $Ca^{2+}$ flux (leak) from the SR via RyRs. However, while RyRs can remain open for many seconds, the fastest gating events are on the millisecond time scale[29]. Therefore, the development of confocal systems with higher temporal and spatial resolution is required before we can measure the kinetics of hotspots of $Ca^{2+}$ flux characterised here with precision and thus confirm whether they truly represent unitary $Ca^{2+}$ release through RyRs. Nevertheless, it is clear that the $Ca^{2+}$ signalling machinery of sub-plasmalemmal, extraperinuclear, perinuclear and nuclear nano-courses incorporates unique receptor components, conferring different nanocourses with the capacity to deliver discrete spatially- and functionally-segregated signals. Supporting the view that the wider network of cytoplasmic nanocourses may represent a circuit for cell-wide communication, LysoTracker Red labelled endolysosomes migrated through this network of cytoplasmic nanocourses (Fig. 2g; 0.25 Hz sampling frequency for dual

labelling; Supplementary Movie 5). By contrast, in these differentiated cells MitoTracker Red labelled mitochondria formed static clusters, as reported previously by others[30], that sat within the nanocourse network (Fig. 2h; Supplementary Movie 6).

**Unloading SR $Ca^{2+}$ into PM-SR junctions relaxes smooth muscle.** We were able to confirm site- and function-specific signalling by employing Maurocalcine, a membrane-traversing peptide from scorpion venom that selectively activates RyR1[31]. Accordingly, Maurocalcine preferentially directed increases in $Ca^{2+}$ flux into subplasmalemmal nanocourses (Fig. 3a–c, e; Supplementary Fig. 5; Supplementary Movie 7), which evoked concomitant myocyte relaxation (Fig. 3d). By contrast there was relatively little change in $Ca^{2+}$ flux within even the most proximal extra/perinuclear nanocourses. We have therefore visualised for the first time unloading of SR $Ca^{2+}$ through RyR1s into the 'superficial buffer barrier' demarcated by PM-SR junctions[32], which confer nanoscale path lengths and have long been predicted to coordinate $Ca^{2+}$ removal from the SR and thus relaxation, as well as SR refilling during prolonged contraction[5,33,34]. This relates clinically[35], confirming that β-adrenoceptors promote pulmonary artery dilation through RyR1-mediated $Ca^{2+}$ release into PM-SR junctions of pulmonary arterial myocytes, for onward removal across the PM through forward mode $Na^+/Ca^{2+}$-exchanger activity[16,17]. Curiously,

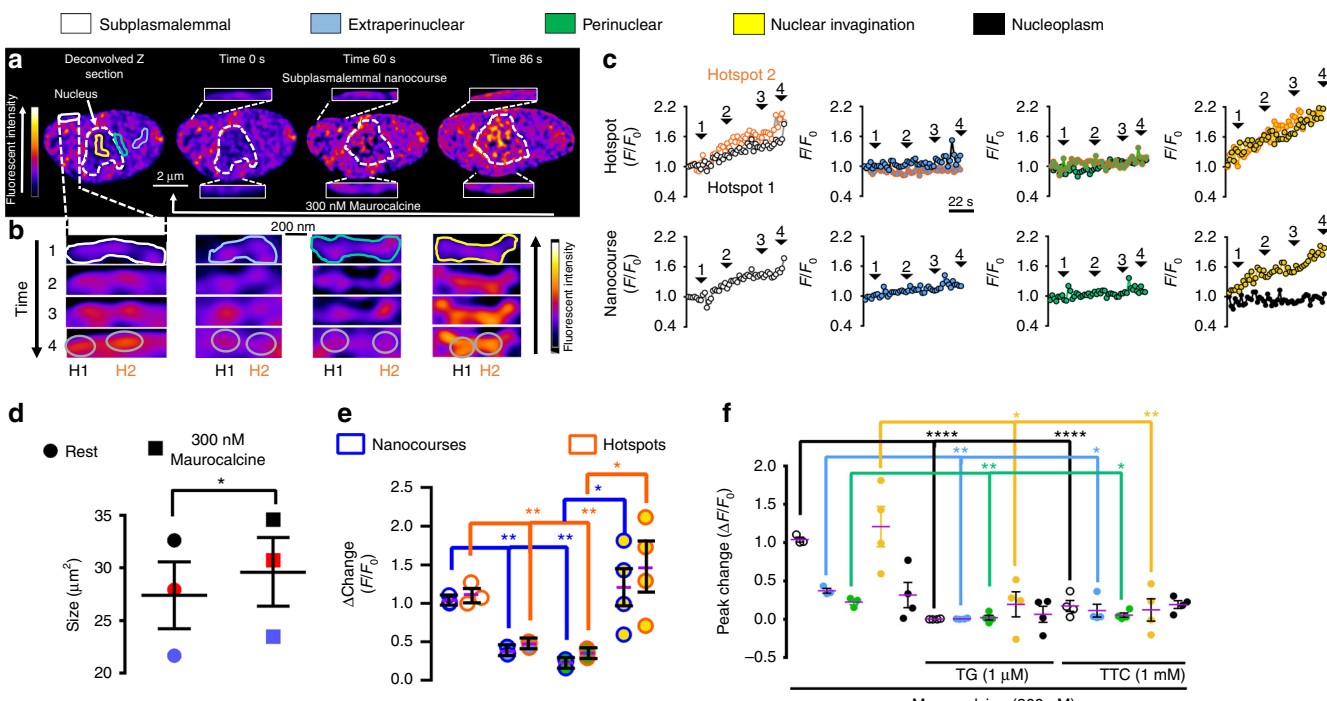

**Fig. 3** Maurocalcine gates $Ca^{2+}$ flux into subplasmalemmal nanocourses and nuclear invaginations. **a** (upper panels) Deconvolved confocal images show pseudocolour representations of Fluo-4 fluorescence intensity in z sections through an acutely isolated arterial myocyte (white broken line identifies nucleus) before and during application of 300 nM Maurocalcine (white arrow). White boxes inset show example subplasmalemmal nanocourses at higher magnification. **b** From left to right, high magnification examples of subplasmalemmal (white), extraperinuclear (blue), perinuclear (green) and nuclear (yellow) nanocourses identified by regions of interest in (**a**), at three different time points. Grey circles identify for each nanocourse, two hotspots (H1, black; H2, orange) of $Ca^{2+}$ flux. **c** Fluo-4 fluorescence ratio ($F_x/F_0$; where $F_0$ = fluorescence at time 0 and $F_x$ = fluorescence at time = x) versus time (sampling frequency = 0.5 Hz) for H1 and H2 of each nanocourse (upper panels, from left to right) compared to the average for the whole nanocourse (lower panels, from left to right). **d** Dot plot shows cell area (μm²; mean ± SEM) before and after extracellular application of 300 nM Maurocalcine ($n = 3$ cells from 3 rats). **e** Dot plot shows peak change ($\Delta F_x/F_0$; mean ± SEM; $n = 3$ cells from 3 rats) for Fluo-4 intensity for hotspots and nanocourses within each region of interest at the peak of the response to Maurocalcine (300 nM). **f** As for (**e**) but for whole nanocourses in the absence and presence of thapsigargin (1 μM, 30 min pre-incubation; $n = 4$ cells from 3 rats) or tetracaine (1 mM; 4 h pre-incubation; $n = 4$ cells from 4 rats); t-test with Welch's correction: *$p < 0.05$, **$p < 0.01$, ***$p < 0.001$, ****$p < 0.0001$. The pseudocolour look up tables in (**a**) and (**b**) indicate relative fluorescence intensity in arbitrary units

however, over the time course of our experiments (2–6 min) Maurocalcine-induced myocyte relaxation was not accompanied by concomitant falls in Ca²⁺ flux into extra/perinuclear nanocourses. If anything, asynchronous Ca²⁺ flux continued within these nanocourses, with perhaps slight increases in activity but no evidence of cell-wide signal propagation (Fig. 3c, Supplementary Movie 7). One explanation for this could be that the relatively small population of RyR1s in extra/perinuclear nanocourses neither face nor couple with the contractile apparatus, but act instead to direct Ca²⁺ flux towards PM-SR junctions via SERCA2b and away from SR release sites occupied by RyR2s/RyR3s that guide myofilament contraction (see below). Consistent with the spatial separation of RyR1s (Fig. 1a–c), Maurocalcine also evoked marked increases in Ca²⁺ flux into nuclear nanocourses adjacent to relatively inactive perinuclear nanocourses, which, therefore, neither generated nor received these signals (Fig. 3a–c, f; Supplementary Fig. 6; Supplementary Movie 7). This exposes the functional segregation of nuclear nanocourses from their nearest neighbour, through the strategic targeting of RyR1s to the ONM that demarcates nuclear nanocourses (see below). It is also evident that Maurocalcine-evoked Ca²⁺ flux within nuclear nanocourses did not propagate freely into the nucleoplasm to any great extent, i.e. Ca²⁺ is released

across the ONM into the cytoplasmic nanocourses defined by each invagination but not directly into the nucleoplasm (Fig. 3a, c, e, f). Consistent with outcomes for nanocourse Ca²⁺ at rest, all responses to Maurocalcine were blocked by tetracaine and by prior depletion of SR stores with thapsigargin (Fig. 3f).

**Nuclear invaginations delimit diverse nanocourse networks.** Using electron microscopy we observed 20–200 nm diameter invaginations, as have others[36], within the nucleus of arterial myocytes in-situ in arterial sections. We could distinguish invaginations of the ONM, forming open transnuclear channels, or shallow, blind invaginations of variable depths reaching into the nucleus (Fig. 4a). As the NE is a double membrane, invaginations also contained inner nuclear membranes (INM) with the luminal space between INM and ONM ranging from 10 to 50 nm. Staining of fixed cells for lamin A, which generally lines the INM, revealed a tubular network formed by nuclear invaginations that criss-crossed the nucleoplasm of these differentiated cells (Fig. 4b), as do nuclear nanocourses. We confirmed that this nucleoplasmic reticulum (NR) held a Ca²⁺ store that was continuous with the perinuclear SR by loading the S/NR lumen using a Ca²⁺ indicator (Calcium Orange) in the absence (Fig. 4c) and presence of SR staining (ER Tracker; Fig. 4d). Therefore,

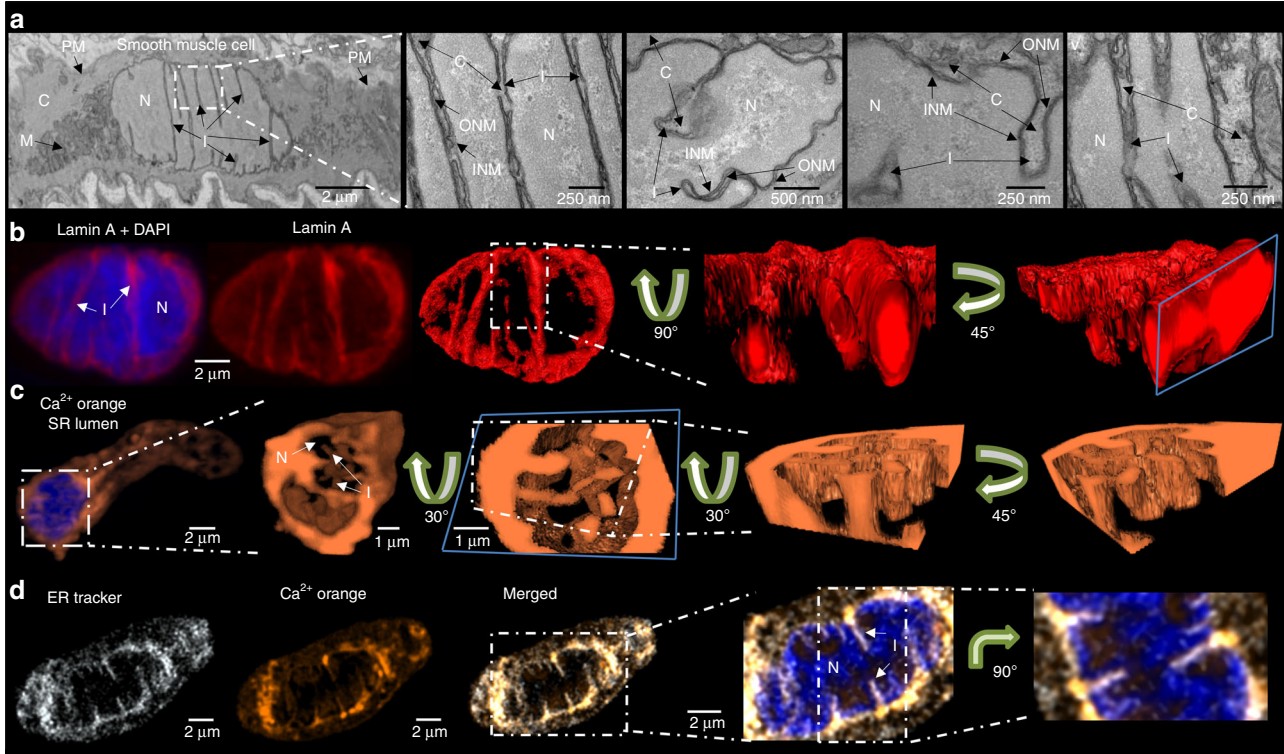

**Fig. 4** Nuclear invaginations demarcate a releasable Ca²⁺ store and cytoplasmic nanotubes. **a** Electron micrographs of artery sections, show (left to right) arterial smooth muscle cells at low and high magnification and identify invaginations (I) of the inner (INM) and outer (ONM) nuclear membrane: PM plasma membrane, C cytoplasm, M mitochondria, N nucleus; confirmed in 4 arteries from 4 rats. **b** Left hand panel shows 3D reconstruction of a deconvolved z stack of confocal images through the nucleus of an arterial myocyte labelled for lamin A (red) with (left panel) and without (middle panel) DAPI (blue) to identify the nucleus (N) and its invaginations (I); confirmed in 54 cells from 14 rats. Right panel, higher threshold and 'digital surface skin' applied to select for nuclear invaginations by way of their higher density of labelling for lamin A. Then, higher magnification transverse section through the 3D image of lamin A labelling shown at 2 different angles. **c** (left to right), 3D reconstruction of a deconvolved z stack of confocal images showing Calcium Orange fluorescence (orange) from within the lumen of the sarcoplasmic (SR) and nucleoplasmic reticulum (SR) of an arterial myocyte, with the nucleoplasm identified (Draq5, blue), higher magnification transverse section through the nucleus of same cell without Draq5 (N, nucleus; I, invaginations), application of digital skin (30° image rotation) and longitudinal section through the centre of the nucleus, then a transverse section through the nucleus (45° image rotation); confirmed in 5 cells from 3 rats. **d** (from left to right), Deconvolved confocal z section through the middle of a pulmonary arterial myocyte showing ER-tracker identified SR and outer nuclear membrane (white), Calcium Orange fluorescence (orange), merged image showing ER-tracker and Calcium Orange fluorescence, higher magnification images with Draq5 identifying the nucleus and its invaginations (N, nucleus; I, invaginations), and a 90° rotation; confirmed in 4 cells from 3 rats

pulmonary arterial myocytes specifically target SERCA1 and RyR1 to the NE in order to facilitate signal segregation within those cytoplasmic nanocourses demarcated by invaginations of the NE.

Closer inspection of Ca²⁺ flux within nuclear nanocourses revealed functional signal segregation in response to not only Maurocalcine (Fig. 3; Supplementary Fig. 6; Supplementary Movie 8) but also to the vasoconstrictor Angiotensin II (Fig. 5a, b; Supplementary Movie 9). Both stimuli triggered increases in Ca²⁺ flux within a subset of nuclear nanocourses, and with distinct spatiotemporal signatures evident in each of these activated nanocourses (Fig. 5b; Supplementary Fig. 6; Supplementary Movies 8 and 9).

The functional reasons for the isolation of nuclear nanocourses are not clear, but it may be to prevent wide-scale gene activation/ inactivation events that could switch cells from a differentiated to proliferative phenotype, operated through specific changes in Ca²⁺ flux. Normal ovoid nuclei tend to have histones carrying both H3K9me2 and H3K9me3 marks, and the chromatin cross-linking protein barrier to auto-integration factor (BAF) associating with NE proteins such as emerin and making the nuclear periphery generally silencing[37–39]. However, interestingly, these marks segregate in differentiated arterial myocytes with the H3K9me2/3 both still at the outer limits of the nucleus but depleted with respect to BAF, and the nuclear invaginations rich with H3K9me2 (Fig. 5c–e) and BAF (Fig. 5f–h) but depleted with respect to H3K9me3 (Supplementary Fig. 7). The combination of H3K9me2/3 together is strongly silencing, but absent the me3 mark and the me2 can reflect a poised state that has been found at myogenic regulators such as the myogenin promoter[40]. It is possible that the non-propagating Ca²⁺ transients in distinct invaginations in some way specifically regulate chromatin in

differentiated cells as the different chromatin marks are concentrated in puncta (Fig. 5c–h): discrete H3K9me2-lamin A puncta (471 ± 38 nm in diameter) were separated by 335 ± 46 nm, while emerin-BAF puncta (361 ± 41 nm in diameter) were separated by 495 ± 61 nm, approximating the 350 nm spacing between the tetracaine-sensitive hotspots of nuclear nanocourses. Potentially Ca²⁺ and/or charge responsive and functionally distinct chromatin domains may therefore be established by nuclear invaginations. Supporting this, qRT-PCR (Fig. 5i) and RNAscope® (Fig. 5j) showed that blocking Ca²⁺ flux through RyRs with tetracaine (1 mM, 90 min pre-incubation) reduced the expression of two genes of interest (identified by RNAseq), one encoding the DNA mismatch repair protein MutL homolog 1 (Mlh1), which can be repressed through interaction with H3K9me2[41,42], and another encoding the S100 calcium binding protein A9 (S100a9), which can be repressed by BAF[43]. That said, further investigation into the role of nuclear invaginations in regulating gene expression will undoubtedly reveal greater complexities of gene regulation, given that individual, acutely isolated smooth muscles possess different types (Supplementary Fig. 8) and different numbers of lamin A and emerin positive invaginations.

**Cell-wide signal propagation and smooth muscle contraction.** As one might expect, increases in Ca²⁺ flux within nuclear invaginations induced by Angiotensin II were accompanied by a Ca²⁺ wave that propagated through all extraperinuclear and perinuclear nanocourses, triggering concomitant myocyte contraction that was evident from reductions in cell surface area (Fig. 6a–c; Supplementary Movie 10; note, when the threshold and $F_{max}$ are set to limit signal saturation, nanocourses are not so clear at rest, see Supplementary Fig. 9). These events were immediately preceded by a rapid fall in Fluo-4 fluorescence intensity within the

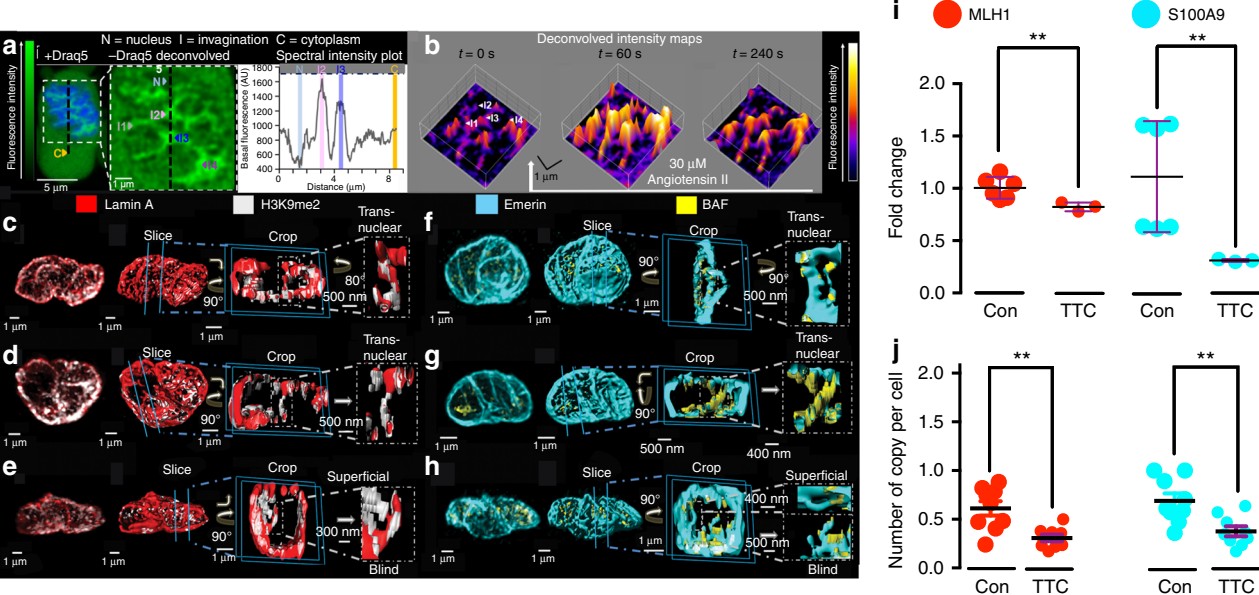

**Fig. 5** Ca²⁺ flux into nuclear invaginations regulates gene expression. **a** (from left to right) Confocal z section of Fluo-4 fluorescence in an arterial myocyte (green) ± nuclear label (blue, Draq5), indicating perinuclear cytoplasm (C), nuclear invaginations (I1, I2, I3, I4) and nucleoplasm (N), and fluorescence intensity plot along vertical dashed black line marked in images. **b** Time series of 3D intensity maps for nuclear region of cell in (**a**) during application of Angiotensin II (30 µM, white arrow). **c** (from left to right) 3D reconstruction of section through the nucleus of a myocyte labelled for lamin A (red; confirmed in 54 cells from 14 rats) and showing co-localisation with H3K9me2 (white; confirmed in 14 cells from 5 rats), then same image with digital skin, sectioned and rotated to identify a transnuclear invagination. **d**, **e** As in (**c**) but different cells. **f–h** As in (**c–e**), but showing BAF co-localisation with emerin; confirmed in 10 cells from 4 rats. **i**, **j** Dot plots show (mean ± SEM) the effect of blocking RyRs with tetracaine (TTC, 1 mM, 90 min pre-incubation) on MLH1 and S100A9 expression in acutely isolated pulmonary arterial myocytes, assessed by **i** q-RT-PCR (assayed in triplicate, for n = 3 rats) and **j** RNAscope (counts per cell, 14–57 cells per plate, n = 9 independent experiments from 3 rats); t-test with Welch's correction: **p < 0.01. The green and pseudocolour look up tables in (**a**) and (**b**) indicate relative fluorescence intensity in arbitrary units

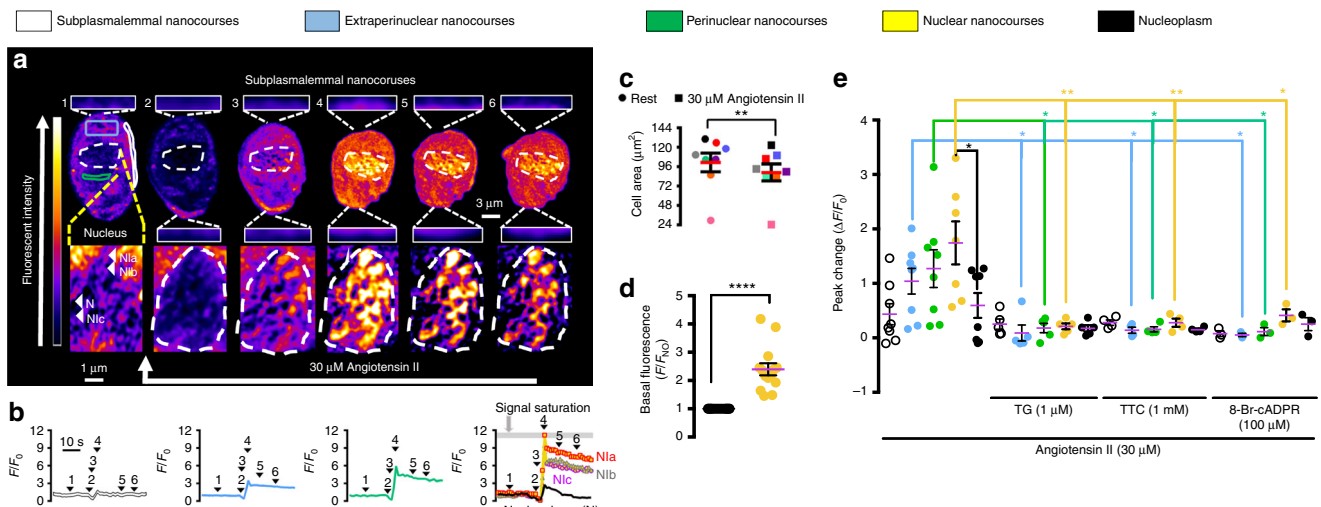

**Fig. 6** Angiotensin II induces myocyte contraction by directing propagating $Ca^{2+}$ signals through nanocourses. **a** (upper panels), Time series of deconvolved z sections shows pseudocolour representations of Fluo-4 fluorescence intensity in an arterial myocyte (white broken line indicates nucleus) during Angiotensin II application (30 μM, white arrow). Insets above and below image show example subplasmalemmal nanocourses at higher magnification. Lower panels as in upper panels but showing changes in Fluo-4 fluorescence intensity within nuclear nanocourses (white arrows indicate: nucleoplasm, N; Nuclear nanocourses, NI1, NI2 and NI3). Note, with cell-wide acquisition of Angiotensin II responses, signal saturation (grey bar) in some nuclear nanocourses was unavoidable (excluded from quantitative analysis). **b** Fluo-4 fluorescence ratio ($F_x/F_0$; where $F_0$ = fluorescene at time 0 and $F_x$ = fluorescence at time = x) versus time (sampling frequency 0.5 Hz) for a subplasmalemmal nanocourse (white), regions of interest within the extraperinuclear (blue) and perinuclear (green) areas of the cell (extra/perinuclear nanocourses could not be followed during contraction), and for nuclear nanocourses (NI1–3, yellow) and nucleoplasm (N, black). **c** Dot plot shows cell area ($\mu m^2$; mean ± SEM) before and after Angiotensin II (30 μM; $n = 8$ cells from 4 rats). **d** Dot plot shows basal Fluo-4 intensity ($F/F_{NO}$; mean ± SEM; $n = 8$ cells from 4 rats) within the nucleoplasm (black) and nuclear nanocourses (yellow). **e** Dot plots show peak change ($\Delta F_x/F_{NO}$; mean ± SEM) for Fluo-4 intensity within specified region of interest after Angiotensin II (30 μM; $n = 8$ cells from 4 rats), in the absence and presence of thapsigargin (1 μM; 30 min pre-incubation; $n = 5$ cells from 5 rats), tetracaine (1 mM; 4 h pre-incubation; $n = 4$ cells from 4 rats) and 8-bromo-cADPR (100 μM; 30 min pre-incubation; $n = 3$ cells from 3 rats); t-test with Welch's correction: *$p < 0.05$, **$p < 0.01$, ***$p < 0.001$, ****$p < 0.0001$. The pseudocolour look up table in (**a**) indicates relative fluorescence intensity in arbitrary units

majority of cytoplasmic nanocourses, except for those at the point of wave initiation, suggesting that Angiotensin II also acts to pre-load the SR with $Ca^{2+}$; perhaps a critical step prior to induction of cell-wide signal propagation. We were unable to study the action of Angiotensin II in the absence of myocyte contraction, because Wortmannin disrupted nanocourse arrays and ML-9 alone did not block cell contraction observed at room temperature. Nevertheless, closer inspection of the $Ca^{2+}$ wave revealed that while marked increases in Fluo-4 fluorescence were recorded in extra/perinuclear regions (Fig. 6a, b, e), there was little or no increase in $Ca^{2+}$ flux into subplasmalemmal nanocourses demarcated by PM-SR junctions (Fig. 6a (upper panels), b, e), which are key to myocyte relaxation (see Fig. 3). This is significant given that RyR2 and RyR3 are preferentially targeted to extra-perinuclear and perinuclear regions, respectively, while RyR1 clusters predominate in subplasmalemmal regions and nuclear invaginations[17,19], because it is RyR2 and RyR3, but not RyR1, that hold the capacity to carry propagating waves by $Ca^{2+}$-induced $Ca^{2+}$ release (CICR)[18,28,44,45]. Irrespective of cellular region, increases in $Ca^{2+}$ flux induced by Angiotensin II were abolished by prior depletion of SR stores with thapsigargin, block of RyRs with tetracaine and by pre-incubation with the cyclic ADP-ribose antagonist 8-bromo-cADPR (Fig. 6d, e), which is in line with the fact that 8-bromo-cADPR also blocks hypoxic pulmonary vasoconstriction at the level of the smooth muscle[46,47]. Accordingly, we have previously shown that intracellular dialysis of high concentrations of cADPR (100 μM) evokes a global $Ca^{2+}$ wave and contraction of acutely isolated pulmonary arterial myocytes[47,48]. Given that cADPR preferentially activates RyR1s and RyR3s[49] but can only sensitise RyR2s to CICR[44], it therefore seems likely that cADPR accumulation within or

proximal to extraperinuclear nanocourses in response to Angiotensin II, serves to activate local subpopulations of RyR1s and/or RyR3s[49] while delivering concomitant sensitisation of RyR2s to `CICR[44], that permits subsequent initiation of a propagating $Ca^{2+}$ signal and thus myocyte contraction. This is in stark contrast to the effect of low concentrations of cADPR (10 μM), the intracellular dialysis of which preferentially releases $Ca^{2+}$ from RyR1s on the peripheral SR proximal to the plasma membrane, to thus evoke membrane hyperpolarization and vasodilation[16,17]. Furthermore, the response to Angiotensin II remained unaffected in the presence of the inositol (1,4,5) trisphosphate receptor (IP$_3$R) antagonist 2APB (not shown). This is in accordance with the fact that IP$_3$Rs do not couple by CICR to RyRs in pulmonary arterial myocytes[50,51], and suggests that this segregation of RyRs from IP$_3$Rs might be conferred by the targeting of RyR2/RyR3s to the SR that demarcates cytoplasmic nanocourses.

That the widths of all extra/perinuclear nanocourses are on the nanoscale (≈500 nm across) is consistent with the finding that the functional $Ca^{2+}$-binding protein calmodulin is tethered proximal to the SR membranes that line myofilament arrays[52], rather than being freely diffusible in the cytoplasm. All relevant path lengths from the SR to myofilaments must therefore be on the nanoscale too. Multiple coordinated actions may thus be delivered by signal segregation between distinct nanocourse networks, enabling nanocourse-specific delivery of $Ca^{2+}$ signals with distinct temporal characteristics (Figs. 3, 5 and 6, Supplementary Fig. 10).

**Remodelling of nanocourse networks upon cell proliferation.** During the transition to proliferating myocytes in culture, the entire, cell-wide network of cytoplasmic nanocourses was

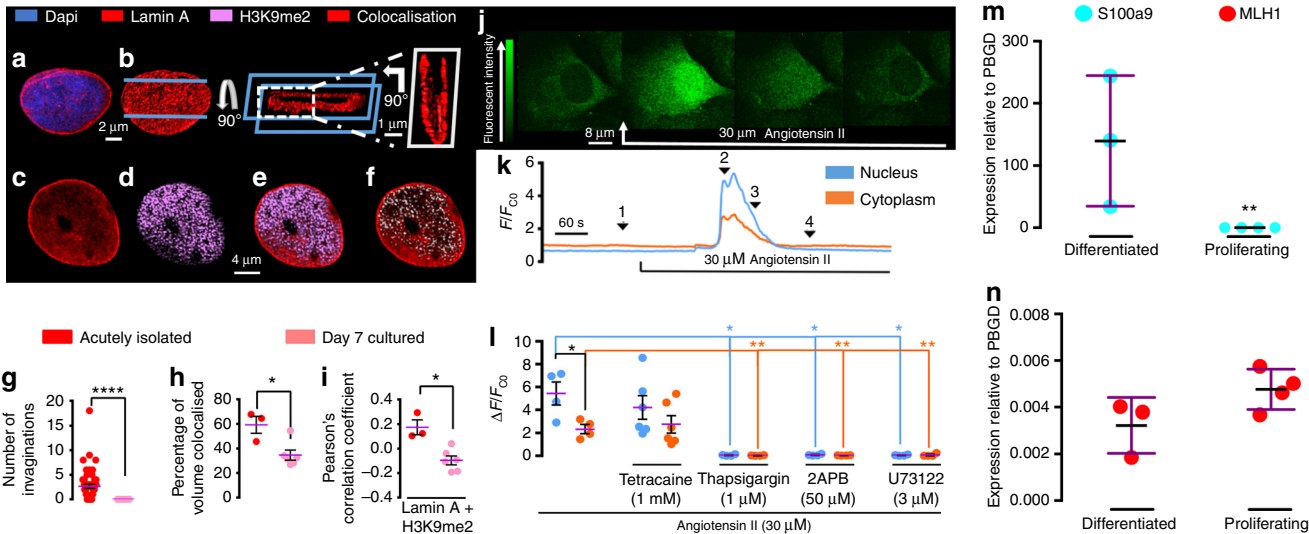

**Fig. 7** Nuclear invaginations and nanocourses are dismantled during myocyte proliferation. **a** (left to right), 3D reconstruction of deconvolved confocal z stack through the nucleus (DAPI, blue) of a proliferating arterial myocyte labelled for lamin A (red). **b** Digital skin applied, a transverse then longitudinal section through and rotations of the nucleus. **c** Different proliferating myocyte. **d** As in (**c**) labelled for H3K9me2 (magenta). **e** (**c**) and (**d**) merged. **f** Merged image showing lamin A (red) labelling and H3K9me2 co-localisation (white). **g** Dot plot shows number of lamin A labelled nuclear invaginations per cell in acutely isolated arterial myocytes (red; $n = 54$ cells from 14 rats) and proliferating cells (pink; $n = 9$ cells from 5 rats). **h** Dot plot shows percentage volume of lamin A and H3K9me2 colocalization ($n = 3$ acutely isolated cells from 3 rats; $n = 6$ cultured cells, from 3 rats). **i** Dot plot shows Pearson's correlation coefficient colocalization ($n = 3$ acutely isolated cells from 3 rats; $n = 6$ cultured cells, from 3 rats). **j** Deconvolved z sections of Fluo-4 fluorescence in a proliferating myocyte (left to right) before and during application of (white arrow) 30 μM Angiotensin II. **k** Records of Fluo-4 fluorescence ($F_x/F_{C0}$; $F_{C0}$ = cytoplasm fluorescence at time = 0, $F_x$ = fluorescence at time = x) against time for the cytoplasm (orange) and nucleus (blue). **l** Dot plot for peak change ($\Delta F_x/F_{C0}$; mean ± SEM; $n = 8$ cells from 4 rats) induced by 30 μM Angiotensin II in the absence and presence of: tetracaine (1 mM; 4 h pre-incubation; $n = 5$ cells from 3 rats); thapsigargin (1 μM; 30 min pre-incubation; $n = 4$ cells from 3 rats), 2-Aminoethoxydiphenyl-borate (2APB, 50 μM; 30 min pre-incubation; $n = 4$ cells from 3 rats) and U73122 (3 μM; 30 min pre-incubation; $n = 4$ cells from 3 rats); one-way ANOVA with Dunnett's multiple comparisons test: *$p < 0.05$, ** $p < 0.01$, ***$p < 0.001$, ****$p < 0.0001$. **m**, **n** Dot plots show (mean ± SEM) q-RT-PCR measures of S100A9 (**m**) and MLH1 (**n**) expression in proliferating myocytes; $n = 3$ rats (in triplicate); one-way ANOVA with Dunnett's multiple comparisons test: **$p < 0.01$

dismantled, inclusive of the rapid loss (≤24 h) of lamin/emerin positive nuclear invaginations (Fig. 7a–i; Supplementary Fig. 11); although in some cells one single lamin/emerin negative, trans-nuclear invagination was identified by ER tracker staining, although these do not appear to support $Ca^{2+}$ signalling (Supplementary Fig. 12). That this phenotypic change is delivered through reconfiguration of the cell-wide nanocourse network that directs $Ca^{2+}$ flux is further highlighted by: (i) A switch in dependency of Angiotensin II-induced $Ca^{2+}$ transients from RyRs to $IP_3Rs$ (Fig. 7j–l); (ii) Unrestricted, cell-wide SR $Ca^{2+}$ release due to loss of cytoplasmic nanocourses; (iii) Loss of the 'nuclear buffer barrier'[53] that opposed direct $Ca^{2+}$ flux into the nucleoplasm in acutely isolated cells (Fig. 7j–l). Accordingly, others have found that myocyte proliferation coincides with whole-scale changes in gene expression inclusive of a decrease in lamin A[54] and RyR expression, and augmented $IP_3R$ expression[55]. Our observations are therefore consistent with the idea that invaginations act to regulate anti-proliferative genes, that is until the proliferative phenotype[56] is ready to be engaged. Unfortunately, rapid loss of nuclear invaginations prevents normal genome manipulations in cultured cells to directly test this proposal. Nevertheless, qRT-PCR showed that during cell proliferation loss of S100A9 expression, but not MLH1 expression, was associated with loss of nuclear invaginations (Fig. 7m, n), consistent with the impact on S100A9 expression of reduced $Ca^{2+}$ flux through RyRs (Fig. 5i, j) and previous reports on S100A9 repression during proliferation of airway smooth muscles[57]. These observations, the distribution of chromatin marks and general tendency of NE-association to keep chromatin repressed[58,59] lends support to the view that NE invaginations

may play a role in genome regulation and cycles of gene repression and activation.

## Discussion

We have identified a cell-wide network of cytoplasmic nanocourses delineated by membrane-membrane nanojunctions of the SR, which provide discrete lines of communication that span the entire cell. Functional specification is therefore determined, as predicted by computer models[60], by the constraints on $Ca^{2+}$ diffusion imposed by SR junctional membranes and by the strategic positioning of different types of $Ca^{2+}$ transporters and release channels targeted to them, through unique kinetics, affinities for $Ca^{2+}$ and mechanisms of regulation[23]. $Ca^{2+}$ flux through RyR1s located within PM-SR nanocourses evoked relaxation with no evidence of cell-wide signal propagation, confirming the existence of a functional 'superficial buffer barrier'[32]. Distinct nanocourses rich in RyR2s and RyR3s carried rapid, propagating $Ca^{2+}$ signals that crossed the entire cell from pole to pole triggering myocyte contraction, yet these signals did not enter those nanocourses demarcated by PM-SR junctions, which constitute the superficial buffer barrier. Therefore, RyR2 subtype designation determines the capacity for rapid signal propagation across specific nanocourses, allowing on the one hand cell-wide synchronous actions as required, but perhaps also conferring the capacity for the onward transfer of a fraction of released $Ca^{2+}$ through SERCA and RyRs that feed functionally distinct subsets of nanocourses, such as nuclear invaginations. Invaginations of the NE confer further segregated and diverse networks of cytoplasmic nanocourses that project deep into the

nucleoplasm, which break down themselves into multiple sub-types based on emerin-BAF and lamin A-H3K9me2 clusters, are served by a unique pairing of ONM resident $Ca^{2+}$ pumps (SERCA1) and release channels (RyR1) and carry $Ca^{2+}$ signals with different spatiotemporal characteristics when activated. That S100A9 expression was attenuated by blocking RyRs and upon loss of nuclear invaginations in culture, strongly supports the view that regulated $Ca^{2+}$ flux across the ONM into nuclear nanocourses might contribute additional levels of genome regulation by, for example, segregating specific chromatin types for cycles of reactivation in differentiated cells and providing a path for related gene repression during phenotypic modulation[56].

Regardless of the functional subdivision of nanocourses observed in pulmonary arterial myocytes, all path lengths from $Ca^{2+}$ release site to targeted signalling complexes must be on the nanoscale, with picolitre volumes of cytoplasm lying within the boundaries of each nanocourse[60]. Relatively small net increases in local $Ca^{2+}$ flux (1–2 ions per picolitre) will therefore be sufficient to raise the local concentration into the affinity ranges of most cytoplasmic $Ca^{2+}$ binding proteins[60]. $Ca^{2+}$ binding proteins may thus operate as local 'switches' that coordinate nanocourse-specific functions, the probability of moving from OFF to ON determined by changes in unitary rather than macroscopic $Ca^{2+}$ flux. Significantly, coincident $Ca^{2+}$ flux can thus be triggered in two distant parts of the cell at the same time, to coordinate, for example, myocyte relaxation and associated gene expression regulation. This draws obvious parallels (Supplementary Fig. 13) to mechanisms of conduction in single-walled carbon nanotubes, which behave as quantum wires that transmit charge carriers through discrete conduction channels, enabling memory, logic and parallel processing. Thus, by analogy, our observations point to the incredible signalling potential that may be afforded by modulating quantum $Ca^{2+}$ flux on the nanoscale, in support of network activities within cells with the capacity to permit stimulus-dependent orchestration of the full panoply of diverse cellular processes. Perhaps more importantly, the cellular intranet conferred by the SR and its associated network activities are not hardwired, reconfiguring to deliver different outputs during phenotypic modulation on the path, for example, to cell proliferation. This in itself suggests that cytoplasmic nanocourses may be common to but vary in nature between different cell types. Supporting this, NE invaginations are a feature of many cell types[10–14] while other junctional complexes of the S/ER vary by cell type and even between different smooth muscles[2,23].

## Methods

**Ethical approval and organ isolation**. All experiments were performed under the United Kingdom Animals (Scientific Procedures) Act 1986. All experiments have complied with all relevant ethical regulations for animal testing and research. Adult male Sprague Dawley rats (~300 g) were sacrificed by cervical dislocation. Skeletal muscle, brain, heart and lungs were removed and placed on ice in physiological salt solution (PSS) of the following composition (mmol/L): 130 NaCl, 5.2 KCl, 1 $MgCl_2$, 1.7 $CaCl_2$, 10 glucose and 10 Hepes, pH 7.4.

**RT-PCR and q-RT-PCR**. For end point PCR, total RNA was extracted from second and third order branches of the pulmonary arterial tree, heart, brain and skeletal muscle using TRIzol® reagent according to the manufacturer's instructions (Invitrogen, UK). Reverse transcription was carried out using 6 μg RNA and 200 U Moloney murine leukaemia virus (Promega, UK) and PCR was performed on 1 μl cDNA with 1 U/μl TAQ DNA polymerase (Biogene, UK). All primer sequences were checked against the GenBank and no cross-reactivity was found. The RT-PCR products over 40 cycles of amplification were resolved by electrophoresis in 1% agarose gels and visualised under UV illumination using an image capture system (Genesnap Image Analysis System, Syngene, UK).

For qPCR RNA from pulmonary arterial smooth muscle was extracted using the High Pure RNA Tissue Kit (Roche) following the manufacturer's guidelines and the concentration determined using the Nanodrop 1000 spectrophotometer (ThermoScientific). cDNA synthesis was carried out using the Transcriptor High Fidelity cDNA synthesis Kit (Roche) following the manufacturers' instructions. For qPCR analysis, 2.5 μl of cDNA in RNase free water was made up to 25 μl with

FastStart Universal SYBR Green Master (ROX, 12.5 μl, Roche), Ultra Pure Water (8 μl, SIGMA) and fwd and rev primers (Origene) for the genes encoding MutL homolog 1 (Mlh1) and S100 calcium binding protein A9 (S100a9). Samples were then centrifuged (13,000g) and 25 μl added to a MicroAmp™ Fast Optical 96-Well Reaction Plate (Greiner bio-one), the reaction plate sealed with an optical adhesive cover (Applied Biosystems) and the plate centrifuged (152g, 2 min). The reaction was then run on a sequence detection system (Applied Biosystems) using AmpliTaq Fast DNA Polymerase, with a 2 min initial step at 50 °C, followed by a 10 min step at 95 °C, then a 15 s step at 95 °C which was repeated 40 times. Then a dissociation stage with a 15 s step at 95 °C followed by 20 s at 60 °C and a 15 s step at 95 °C. Data were normalised to a housekeeping gene (Ipo8, acutely isolated, differentiated cells; Pdgb, cultured cells). Negative controls included control cell aspirants for which no reverse transcriptase was added, and aspiration of extracellular medium and PCR controls. None of the controls produced any detectable amplicon, ruling out genomic or other contamination.

Bespoke primers used were as follows:

SERCA1 Sense CTC ACT TCC AGT CAT CGG GCT AG (nucleotide position: 3084–3106), predicted product size 340 bp, Antisense GTC AGC TAG TTG CCT TGT CCC TG (nucleotide position: 3423–3401); SERCA2a Sense CCT GTC CAG TTA CTC TGG GTC (nucleotide position: 2569–2589), predicted product size 694 bp, Antisense GCT AAC AAC GCA CAT GCA CGC (nucleotide position: 3262–3242); SERCA2b Sense GCC AAC ATT GCC TAT TCA GTG GCA C (nucleotide position: 4382–4406) predicted product size 501 bp, Antisense GGA TGT TGG AAG GCA TTG GAG G (nucleotide position: 4882–4861); SERCA3 Sense CTC WGA GAA CCA GTC ACT GCT GCa (nucleotide position: 2879–2901) predicted product size 264 bp, Antisense GCT CYC AGG ATT TAC TTC AGG TCCb (nucleotide position: 3142–3119). Primer sequences were checked against the GenBankTM, and no cross-reactivity was found.

Commercial primers used were as follows:

Rat s100a9 primer PrimePCR™ SYBR® Green Assay (BioRad; catalogue number: 10025636). Mlh1 primer, Rat, Unique Assay ID: qRnoCID0007985 (BioRad; catalogue number: 10025636); Rat lamin A primer Rn_Lmna_1_SG QuantiTect Primer Assay (Qiagen; catalogue number: QT00192255); PBGD primer PCR SYBR Green Assay (BioRad; catalogue number: 10025220). Ipo8 Quantitect Primer Assay (Qiagen; catalogue number: QT01057518).

**Western blotting**. Tissues samples were rapidly frozen in liquid nitrogen. Small segments of tissue were ground to a powder under liquid nitrogen and homogenised in an appropriate volume of ice-cold buffer (mmol/L; 50 Tris, 150 NaCl, 50 NaF, 5 Na pyrophosphate, 1 EDTA, 1 EGTA, 1 DTT, 0.1 benzamidine, 0.1 PMSF, 0.2 mannitol, 0.1% (v/v) Triton, pH 7.4) using a motor-driven pestle. The homogenate was kept on ice for 30 min and then centrifuged (13,000g, 5 min, 4 °C). The supernatants were removed and their protein concentrations determined.

Equal amounts of the nuclear fraction (free of nuclear and whole cell debris) from second order pulmonary artery smooth muscle and mircrosomal fractions of lysates from second order pulmonary arterial smooth muscle, brain and skeletal muscle homogenates were separated on 10% bis-acrylamide gels (NuPAGE™ electrophoresis system; Invitrogen, UK) and blotted onto nitrocellulose membranes. Blots were probed with sequence-specific antibodies for SERCA1 (1:2500, mouse monoclonal, raised against residues between 199 and 505 of rabbit SERCA1, Abcam, UK). Every blot was also probed with a polyclonal antibody for actin to ensure equal protein loading (Sigma, UK) alongside the SERCA antibodies. Detection was performed with horseradish peroxidase-conjugated secondary antibodies using the ECL system (GE Healthcare, UK).

**Electron microscopy**. Secondary and tertiary branches of the pulmonary artery were dissected and immediately immersed in fixative solution, 4 °C. The primary fixative solution contained 2.5% glutaraldehyde + 4% paraformaldehyde (Ted Pella, Redding, CA, USA) in 0.1 M sodium cacodylate buffer (Canemco & Marivac, Gore, QC, Canada). The artery rings were then washed five times (30 min per wash) in 0.1 M sodium cacodylate and left overnight at, 4 °C. In the process of secondary fixation, the tissue rings were fixed with 1% $OsO_4$ + 3% $C_6FeK_4N_6{\cdot}3H_2O$ (Ted Pella, Redding, CA, USA) in 0.1 M sodium cacodylate buffer for 1 h at 18–24 °C followed by three 10 min washes with distilled water. The samples were then dehydrated in increasing concentrations of acetone (30, 50, 70, 80, 90, and 95%; 15 min each). In the final process of dehydration, the samples underwent 3 washes in 100% acetone. The artery rings were then resin-infiltrated in increasing concentrations (25, 50, and 75% in acetone) of TAAB 812 resin mix (TAAB Laboratories Equipment Ltd., UK) and left in pure resin overnight in obscurity. The blocks were finally resin-embedded in moulds and polymerised in an oven at 60 °C for 8 h. For standard (2D) electron microscopy imaging, 80-nm sections were cut from the embedded sample blocks on a Reichert Ultracut-E microtome (Leica Microsystems, Vienna) using a diamond knife (Diatome, Biel, Switzerland) and were collected on uncoated 100- and 200-mesh copper grids (hexagonal or square meshes; Ted Pella, Redding, CA, USA). The sections were post-stained with 1% uranyl acetate (Canemco & Marivac, Gore, QC, Canada; pH not recorded) and Reynolds lead citrate (Fisher Scientific Company, USA; pH not recorded) for 3 and 4 min, respectively. Electron micrographs at various magnifications were obtained with a Jeol JEM-1010 high resolution transmission electronic microscope (JEOL, Tokyo, Japan).

**Smooth muscle cell isolation.** Single arterial smooth muscle cells were isolated from secondary and tertiary order branches of the pulmonary arterial tree. Briefly, arteries were dissected out and placed in low $Ca^{2+}$ solution of the following composition (mM): 110 NaCl, 5 KCl, 2 $MgCl_2$, 0.5 $NaH_2PO_4$, 0.5 $KH_2PO$, 15 $NaHCO_3$, 0.16 $CaCl_2$, 0.5 EDTA, 10 glucose, 10 taurine and 10 Hepes, pH 7.4. After 10 min the arteries were placed in the same solution containing 0.5 mg/ml papain and 1 mg/ml bovine serum albumin and kept at 4 °C overnight. The following day 0.2 mmol/L 1,4-dithio-DL-threitol was added to the solution, to activate the protease, and the preparation was incubated for 1 h at room temperature (22 °C). The tissue was then washed 3× in fresh low $Ca^{2+}$ solution without enzymes, and single smooth muscle cells were isolated by gentle trituration with a fire-polished Pasteur pipette. Cells were stored in suspension at 4 °C until required.

**Smooth muscle cell culture.** For $Ca^{2+}$ imaging by confocal microscopy, freshly isolated pulmonary arterial smooth muscle cells were suspended in nominally $Ca^{2+}$ free solution with a cell density of 1000,000–2000,000 cells per ml. 25 µl of cell suspension was plated on Fluorodishes (WPI, USA) and cultured for 7 days. For immunocytochemistry, the same amount of cell suspension was added to a 13 mm circular coverslip sat in a 12-well plate. Cells were allowed to adhere for 45 min at room temperature in a cell culture hood. 2 ml of culture medium of following composition (44.5% Waymouth's Medium, 44.5% Ham's F12 Nutrient Mixture, 10% foetal bovine serum (FBS), 1% Penicillin-Streptomycin) was then added to each Fluorodish or each well of the 12-well plate and the cells were incubated at 37 °C with air + 5% $CO_2$. FBS was added to enhance cell growth and proliferation[61], while antibiotics were added to reduce the chance of infection. The culture medium was replaced every 3 days for the best outcomes. Proliferating cells were identified by shape, morphology and reductions in myosin heavy chain labelling[62].

**Immunocytochemistry.** Cells were placed onto poly-D-lysine-coated coverslips, fixed using ice-cold methanol for 15 min at −20 ºC, permeabilized by three 5 min washes with 0.6% Triton X-100 in phosphate-buffered saline (pH 7.4) followed by 20 min wash with blocking solution (1% bovine serum albumin, 4% goat serum, 10% donkey serum and 0.3% Triton X-100 in phosphate-buffered saline, pH 7.4). The cells were incubated overnight at 4 °C with the sequence-specific antibodies for SERCA1 (1:500), SERCA2a (1:500) and SERCA2b (1:500). In addition, paired samples of cells were incubated overnight at 4 °C with affinity-purified rabbit anti-RyR1 (1:500; raised against the peptide residues 4476–4486[63]), anti-RyR2 (1:500; raised against the peptide residues 1344–1365[64]) and anti-RyR3 (1:500; raised against the peptide residues 4236–4336[65]). To visualise trans-nuclear membrane proteins, cells were incubated overnight at 4 °C with the sequence-specific antibodies for A-type lamin (1:200), emerin (1:200), H3K9me2 (1:200), BAF (1:50). Coverslips were washed four times with blocking solution and incubated with goat anti-mouse Alexa448-conjugated secondary antibodies (1:200; excitation 490 nm, emission 518 nm), donkey anti-mouse IgG (H + L) highly cross-adsorbed (1:200; excitation 490 nm, emission 518 nm), Alexa Fluor® 546 goat anti-mouse IgG (H + L), highly cross-adsorbed, goat anti-rabbit Texas Red-conjugated secondary antibody (1:200; excitation 596 nm, emission 620 nm), donkey anti-rabbit IgG (H + L) highly cross-adsorbed secondary antibody (1:200; excitation 596 nm, emission 620 nm), or donkey anti-goat IgG (H + L) cross-adsorbed secondary antibody (1:200; excitation 650 nm, emission 668 nm). To visualise nuclei the coverslips were incubated with 4-,6-diamidino-2-phenylindole (DAPI, 1 µg/ml; excitation 358 nm, emission 461 nm) for 15 min at room temperature. Then, the coverslips were washed 3 times with phosphate-buffered saline and 2 times with phosphate-buffered saline containing 0.1% Tween-20. After 5 min of air drying the coverslips were attached to slides by anti-fading mountant (2.4 g Mowiol 4–88, 6 g of glycerol, 2 ml of 0.2 M Tris–HCl, pH 8.5, 2.5% 1,4 diazabicyclo (2.2.2.) octane). For controls, the primary antibody was omitted.

Images were acquired and processed by either:

(1) A Deltavision imaging system (Applied Precision, UK) consisting of an Olympus IX70 inverted microscope with an Olympus PlanApo 60×, 1.40 n.a. oil immersion objective and a Photometric CH300 charge-coupled device camera. Z section (0.2 µm) stacks were taken through cells. Images were deconvolved using Softworx acquisition and analysis software (Applied Precision, UK).

(2) A Nikon A1R + confocal system via a Nikon Eclipse Ti inverted microscope with a Nikon Apo 63× λS DIC N2, 1.25 n.a. oil immersion objective (Nikon Instruments Europe BV, Netherlands). Images were deconvolved, with pinhole at 0.7 Airy units, and pixel settings of 60 nm in $x,y$ and 150 nm in $z$. Estimated resolution with these settings is $x,y = 170$ nm, $z = 540$ nm. Image processing and 3D rendering was carried out using Imaris (Bitplane, Oxford Instruments, UK). Images were deconvolved using Huygens Essential (Scientific Volume Imaging, Netherlands).

Primary antibodies used were as follows:

SERCA antibodies—SERCA1, mouse monoclonal, raised against residues 199–505 of rabbit SERCA1 (Abcam). SERCA2a, rabbit polyclonal, raised against residues 989–997 of pig SERCA2a[17], a kind gift from F. Wuytack, University of Leuven, Belgium. SERCA2b, rabbit polyclonal, raised against residues 1032–1043 of pig SERCA2b[17], a kind gift from F. Wuytack, University of Leuven, Belgium. SERCA3, rabbit polyclonal, raised against residues 29–39 of mouse SERCA3 (Abcam).

RyR antibodies—RyR1, affinity-purified rabbit anti-RyR1 raised against the peptide residues 4476–4486[63]. RyR2, anti-RyR2 raised against the peptide residues 1344–1365[64]. RyR3, anti-RyR3 raised against the peptide residues 4236–4336[65]. All were a kind gift from S. Fleicher, Vanderbilt University, TN, USA.

NE proteins—Emerin polyclonal antibody were a kind gift from Glenn Morris and outcomes confirmed by use of commercial emerin polyclonal antibodies (Invitrogen, catalogue number: PA5–51424). Thoroughly characterised, see http://www.glennmorris.org.uk/mabs/WCIND.htm.

Lamin A polyclonal antibody (ABCAM; catalogue number: ab26300), specificity previously confirmed[66].

Histone H3 di methyl K9 (ABCAM; catalogue number: ab194680). Di-Methyl-Histone H3 Lys9 polyclonal antibody (Invitrogen; catalogue number: PA5–16195). Epigenetic mark antibodies were selected by loss of labelling in yeast with enzymes that confer the marks deleted.

Secondary antibodies used were as follows:

Texas Red-conjugated goat anti-rabbit secondary (Jackson ImmunoResearch, USA; catalogue number: 200-072-211).

Alexa Fluor® 488-AffiniPure goat anti-rabbit IgG (H + L) (Stratech Scientific Limited for Jackson ImmunoResearch; catalogue number: 111-545-144-JIR).

Alexa Fluor® 546 goat anti-mouse IgG (H + L), highly cross-adsorbed (Life Technologies; catalogue number: A-11030).

Alexa Fluor® 488 donkey anti-mouse IgG (H + L) highly cross-adsorbed (Invitrogen; catalogue number: A-21202).

Alexa Fluor 568 donkey anti-rabbit IgG (H + L) highly cross-adsorbed, (Invitrogen; catalogue number: A10042).

**Semi-quantitative analysis of fluorescent labelling.** 3D rendered images of pulmonary arterial smooth muscle cells were obtained using Volocity software (Perkin-Elmer, UK) and subdivided into three defined volumes that excluded the DAPI labelled nucleus, namely the perinuclear (the area of the cell located within 1.5 µm of the DAPI labelled nucleus), the subplasmalemmal (the area of the cell located within 1.5 µm of the plasma membrane) and the extra-perinuclear regions (the remaining volume of the cell). The volumes occupied by these three defined regions were measured and then the density of fluorescent labelling in that region determined by dividing the volume of protein labelling (e.g. for SERCA2a) within a given region by the volume of that region as described in detail previously[18]. Thus, data are presented as the mean ± SEM of the volume of labelling per µm3 of a given region ($\mu m^3$ per $\mu m^3$).

**Confocal $Ca^{2+}$ imaging.** The $Ca^{2+}$-loaded sarcoplasmic reticulum of acutely isolated cells was identified by incubation (30 min) with 5 µM Calcium Orange (excitation 549 nm; emission 576 nm) and/or 1µM ER-tracker (excitation 374 nm; emission 430 nm) at 37 °C, washed 5 times with indicator-free PSS and allowed to equilibrate for at least 45 min at 18–22 °C prior to experimentation.

For cytoplasmic calcium, cells were incubated for 30 min with 5 µM Fluo-4-AM (excitation 494 nm; emission 506 nm) in PSS at 18–22 °C, and then washed 5 times with Fluo4-AM free PSS and allowed to equilibrate for at least 45 min prior to experimentation. Intracellular $Ca^{2+}$ concentration was reported by Fluo4 fluorescence.

In each case the Fluo-4 fluorescence ratio was recorded at 22 °C with a sampling frequency of 0.5 Hz, using a Nikon A1R + confocal system via a Nikon Eclipse Ti inverted microscope with a Nikon Apo 40 × λS DIC N2, 1.25 n.a. water immersion objective (Nikon Instruments Europe BV, Netherlands). Draq 5 (excitation 646 nm; emission 681 nm) was used to identify the nucleus. Experiments were processed with ImageJ software (Rasband WS. ImageJ, U.S. National Institutes of Health, Bethesda, MD, USA, imagej.nih.gov/ij/, 1997–2012). Note, each image file was assessed manually to insure accuracy and consistency of measurement within the regions of interest specified. Images were acquired using the Galvano scanner (400 Hz). Laser power was < 3% (offset = 0), Hv ≈120 V, image size 1024 × 1024 (16 bit), frame time = 2 s, sampling frequency 0.47 Hz (1.1 frames s⁻¹). Note: greater spatial resolution is afforded by the Nikon A1R + due, in part, to the fact that the integration time is equal to the pixel time (for further information see: https://m.nikoninstruments.com/content/download/15419/…/file/A1_2CE-SCNH-4r.pdf; https://igmm-imaging.squarespace.com/s/Nikon-A1R-User-Manual-web.pdf). Deconvolution was completed with pinhole at 0.7 Airy units, and pixel settings of 60 nm in $x,y$ and 150 nm in $z$. After deconvolution under these conditions, (488 nm excitation, 2D) the estimated resolution is $XY = 230$ nm and $Z = 582$ nm. We excluded from analysis any cell that exhibited contraction or other movement artefact sufficient to alter the focal plane and/or compromise our ability to take reliable measurements of regions of interest. Images were deconvolved using Huygens Essential (Scientific Volume Imaging, Netherlands).

**RNAscope.** RNAscope®[67] was completed according to manufacturer's (Advanced cell diagnostics) instructions, described in the RNAscope® Fluorescent Multiplex Kit Manual. All probes were supplied by Advanced Cell Diagnostics Srl. Via Calabria, 15 20090 Segrate (Milan), Italy: RNAscope probe for Rat Mlh1 ID; 81685 (catalogue number: 300031-C2); RNAscope probe for Rat S100a9 ID; 94195 (catalogue number: 300031-C3).

**Data presentation and statistical analysis**. Data are presented as the mean ± SEM. For data shown in Fig. 1 comparisons between the groups were carried out in MINITAB 14 (MINITAB, LLC, USA) using one-way ANOVA followed by a Tukey post-hoc test. Probability values less than 0.05 were considered to be statistically significant.

For data shown in Figs. 2–6, comparisons between the groups were carried out using GraphPad Prism (GraphPad Software Inc., USA) one-way ANOVA followed by Dunnett's multiple comparisons test and using t-test with Welch's correction. Similar levels of significance were obtained with each test for all data sets. Outcomes presented are for the statistical test considered most appropriate for the data set in question. Probability values less than 0.05 were considered to be statistically significant.

**Drugs and chemicals**. Fluo-4, AM, cell permeant (Molecular Probes®) 10 × 50 μg (Life Technologies; catalogue number: F-14201). Calcium Orange (Thermo Fisher; catalogue number: C3015). DRAQ5™ Fluorescent Probe Solution (5 mM; Thermo Fisher; catalogue number: 62251). LysoTracker™ Red DND-99—Special Packaging (Thermo Fisher; catalogue number: L7528). ER-Tracker™ Blue-White DPX, for live-cell imaging (Thermo Fisher; catalogue number: E12353). MitoTracker™ Green FM (Invitrogen; catalogue number: M7514). Thapsigargin (ABCAM; catalogue number: ab120286); Maurocalcine (ABCAM; catalogue number: ab141860). Angiotensin II (ABCAM; cataologue number: ab120183); Tetracaine (Sigma-Aldrich catalogue number: T7383).

**Reporting summary**. Further information on research design is available in the Nature Research Reporting Summary linked to this article.

## Data availability
The data sets generated during and/or analysed during the current study are available from the corresponding author on reasonable request.

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

## Acknowledgements

Funding for this work was from a U.K. British Heart Foundation Programmed Grant (RG/12/14/29885; renewal rejected, December 2018) which supported the work of J.N.-D., J.D. and A.M.E. Microscopy was done in the IMPACT Imaging Facility at the University of Edinburgh. The work of J.D. also received support from the China Scholarship Council (201508060127). J.H.C. (PG/03/065) and N.P.K. (PG/05/128/19884) were supported by British Heart Foundation project grants to A.M.E. that expired in 2006 and 2007. E.C.S. and P.M. were supported by Wellcome grant 095209 and the Wellcome Centre for Cell Biology core grant 092076. J.D. and A.M.E. thank Heather McClafferty for support with respect to qPCR. The authors thank Sidney Fleischer, Frank Wuytack and Glenn Morris for the kind gift of antibodies. Finally, A.M.E. thanks Gordon Murray, Barclay Thomson and other members of the shortest day club for enlightening discussions on quantum tunnelling, and what might be possible when all the ducks are in a row.

## Author contributions

A.M.E. conceived of this study and wrote the manuscript. J.D and E.C.S. provided detailed feedback on the manuscript. A.M.E., J.D. and J.N.-D. designed experiments. J.H.C., N.P.K. and A.M.E. performed immunocytochemistry and deconvolution microscopy on RyR and SERCA subtypes between 2004 and 2006. J.H.C. and A.M.E. performed RT-PCR and western blots in 2005. J.N.-D. performed electron microscopy. J.D., J.N.-D. and A.M.E. performed confocal microscopy, calcium imaging, immunocytochemistry, deconvolution and associated data analysis. J.D. and A.M.E. carried out RNAscope and qRT-PCR. E.C.S. provided invaluable advice regarding the study of lamin A, H3K9me2/3, emerin and BAF. P.M. and E.C.S. provided materials and advice on immunocytochemistry relating to double labelling for nuclear envelope associated proteins. All authors discussed the results and provided feedback with respect to their contribution to the manuscript.

## Additional information

**Competing interests:** The authors declare no competing interests.

