## [Peer Review File · Nature Communications]

Reviewers' comments:

Reviewer #1 (Remarks to the Author):

This MS puts forward a new mechanism for Ca²⁺ signalling in cells. The authors suggest the existence of nanocourses insulated from the rest of the cytosol and nucleoplasm that are specialised pathways for Ca signalling to its effectors.

The ideas put forward are very interesting and could be of great significance.

The MS relies heavily upon deconvolution of Ca²⁺ imaging and high resolution/deconvolution of immunostaining of the players involved in Ca²⁺ handling to reach its conclusions.

Pharmacological interventions are employed that strongly support the reports that different RyR isoforms are involved in mediating nuclear vs subplasmalemmal Ca²⁺ release events and that a replete SR is required for generation of Ca²⁺ changes.

While interesting, I don't consider that there is enough direct evidence to support the conclusions drawn.

Cells are densely packed with organelles that limit diffusion of Ca²⁺. But despite this, Ca²⁺ changes are reported to be global in nature. These organelles are also constantly moving - this is well established for the ER and for mitochondria. As such the shape of the regions of the cytosol devoid of these organelles is also changing. These shape changes would also impact on the local concentration of the Ca²⁺ indicator dye employed. Together, these processes could therefore result in similar observations as described in this MS.

Given the role of the cytoskeleton in mediating the movement of the ER (actin) and mitochondria (microtubules), it would be of interest therefore to prevent movement of these organelles.

The MS reports hotspots of Ca²⁺ that are presumably generated from clusters of RyRs. these events should be analogous to Ca²⁺ sparks. Given that Ca²⁺ sparks have a defined lifetime and dimensions, more efforts should be made to capture these events using high speed imaging approaches. The direct measurement of Ca²⁺ sparks would strengthen the proposal that hotspots are bona fide Ca²⁺ release events.

Imaging at higher temporal resolution is absolutely required to determine whether these release events propagate and truly act as a signalling pathway.

Fluo 4 signals are not calibrated and response to Ca is known to be affected by concentration and local environment. Does the Ca²⁺ indicator report Ca²⁺ similarly in all types of hotspots. It is well established that the environment of the nucleus affects the Ca²⁺ dependent changes in fluo 4 - for e.g. basal fluorescence is higher than in the cytosol. Could the authors assess the dynamic range and Ca²⁺ sensitivity of their indicator in the hotspots analysed.

To compensate for local concentration differences of the dye, an approach were a second non Ca sensitive dye is co-loaded may help.

It is not clear to me at present whether these hotspots are stochastic release events with no real function or have a true function. The authors provide correlation of the nanocourse with location of the epigenetic marks H3K9me2 and H3K9me3, which are known to be enriched under the nuclear envelope and mediate gene repression. Genes covered by H3K9me3 are particularly associated with heterochromatin and permanently silenced in differentiated cell types. While the observations are interesting, there is no link between Ca²⁺ change and dynamic alteration of the abundance of these marks or activation of gene expression.

The authors propose that the effects of AngII are mediated via cADPR and RyRs. while cADPR can directly activate RyR1, it only acts to sensitise RyR2. Based on this differential sensitivity, the authors propose the differential roles of RyR1 and 2 in mediating Ca changes in different cellular compartments.

RyRs open with a defined open probability - they are not permanently closed but undergo stochastic openings in a manner that is increased by Ca. while the authors suggest that the reason why RyR2 are not activated by AngII is because cADPR does not directly activate them. However, if these channels are undergoing spontaneous release events, wouldn't cADPR not act to enhance channel coupling subsequent to a stochastic opening thereby generating a hotspot?

This effect of cADPR could also be affected by the density of RyR expression. Recent evidence from cardiac myocytes - modelling and imaging - has shown a key role for the size of the cluster and its organisation in generating Ca²⁺ sparks. The more organised the cluster, the more efficient it is in generating sparks. As such, larger clusters with more spontaneous openings could be more sensitive to cADPR than smaller disorganised clusters. Could the authors therefore examine RyR clusters dimensions in their cells.

It would be useful to probe the effect of Ca²⁺ buffering with BAPTA on the proposed differential activation of RyR1 and 2. IF directly activated by cADPR, BAPTA should be without effect whereas activation of channels/hotspots by CICR would be reduced.

A similar effect of cADPR could also be brought about by IP3 induced Ca release generated downstream of AngII. This effect of IP3 would not need to be great to sensitise RyRs and bring about a Ca²⁺ change. While the authors may have probed the role of IP3 in other work and in nuclear Ca regulation in this MS, the authors should also test the role of IP3 in the Ca²⁺ changes in the nanocourses reported in this MS.

The authors indicate that the hotspots asynchronously activate. What is the reason for this? As Ca²⁺ can promote its own release, I would think it likely that the nanocourses should also facilitate coupling between RyR clusters (that underlie the hotspots). The temporal resolution of imaging is clearly substantially too slow to detect all release events if these hotspots are arising from a cluster of RyRs in a manner analogous to sparks. Can the authors also rule out the possibility that the Ca²⁺ indicators used are also not acting to restrict diffusion of the Ca²⁺ release events. As suggested above, the concentration of the indicator may be substantially higher in the nanocourses than what is usually observed in the bulk cytosol.

These observations have all been made in smooth muscle cells and it is interesting that effects are lost on culture. Could the authors provide information whether the nanocourses are a smooth muscle specific phenomenon or are they of wider importance. If not could the authors explicitly state that the nanocourses are a particular feature of smooth muscle cells.

The quality of image capture is key to this study. In the absence of sufficient image resolution or over iteration in deconvolution, regions of fluorescence can be blurred to an extent that they merge. Could the authors therefore provide further information regarding their parameters of image capture - specifically, pixel size in x, y and z. this is required for the Ca imaging - I presume with the Nikon A1R and also for the immunofluorescence analysis using the Deltavision system.

For the Ca²⁺ imaging was a z stack collected and then deconvolved to give a slice or is deconvolution in 2D only?

Given the z resolution of the system is likely closer to 600 nm, the authors cannot rule out Ca hotspots, fluorescence changes arising from under the plane of focus.

Could the authors clearly indicate which images are captured with which system in the figure legends.

Minor:

1. Page 1 of the introduction the authors state: 'Evidence suggests that within the sarcoplasmic reticulum (SR), the primary intracellular Ca²⁺ store, multiple releasable pools of Ca²⁺ exist'. Is this really true? Or is it that Ca²⁺ can be released by a number of different mechanisms. In other cell systems, it has been shown that the ER is a contiguous organelle.
2. Page 4. A space should always be present between unit and integer.
3. Different SERCA isoforms are reported in different subcellular compartments. What would be the reason for this - what could be the benefit? As different SERCA isoforms have different Ca affinities and transport rates, different rates of recovery of the Ca²⁺ signals may be observed at different cellular locations. Could the authors please examine.

Reviewer #2 (Remarks to the Author):

Duan et al. extensively examined arterial smooth muscle cells using various techniques including state-of-the-art approaches, and report the distinct residency and functional role of RyR subtypes. Their observations are definitely novel and likely make a big deal, which would fundamentally elevate our understanding of cellular Ca²⁺ signaling in smooth muscle. However, several improvements may be still needed.

1. Are the home-made antibodies against RyRs sure in subtype specificity? This point must be very important because the results largely depend on the subtype-specific antibodies. For example, can the authors present WB and IHC data in skeletal and heart muscles? It has been already established that skeletal muscle contains abundant RyR1 and also a small amount of RyR3, while cardiac muscle specifically contains RyR2.
2. Both RyR2 and RyR3 have been frequently examined in smooth muscle, however, RyR1 was not. Is it sure that arterial muscle contains all RyR subtypes? Which subtype expression is altered during the proliferating stage? To re-confirm such points, RT-PCR experiments using subtype-specific primers are probably useful.
3. After publication of this study, without reservation, many researchers in the Ca²⁺ signaling field will focus on the molecular mechanisms underlying the RyR subtype-specific characteristics, such as subcellular localization and functional channel crosstalk in smooth muscle. The authors can propose and explain such mechanisms in the Discussion part?
4. By using commercial microscopes, optical resolution is limited and deconvolution technique has been recently developed. Deconvolution seems to work well in previously reported cases, while inappropriate deconvolution should mess up observations. Therefore, the procedure for deconvolution is critical to reproduce the results and to convince general readers. The basic procedure of the deconvolution utilized should be briefly described in the Method section.
5. "Quantum tunnelling" is a provocative term, but it seems to be a defined technical term in the quantum mechanics of physics. If the authors simply use it as paraphrase of "Ca²⁺ tunnelling", "quantum" cannot be appropriate here. In addition, "calcium tunnels" has been previously proposed by Petersen et al. (Cell Calcium 42, 373-378, 2007). To distinguish the reported concept from the Petersen's proposal, the authors may need other words in this study.

Rebuttal to Reviewers Comments

Reviewer #1 (Remarks to the Author):

This MS puts forward a new mechanism for Ca²⁺ signalling in cells. The authors suggest the existence of nanocourses insulated from the rest of the cytosol and nucleoplasm that are specialised pathways for Ca signalling to its effectors.

The ideas put forward are very interesting and could be of great significance.

The MS relies heavily upon deconvolution of Ca²⁺ imaging and high resolution/deconvolution of immunostaining of the players involved in Ca²⁺ handling to reach its conclusions.

Pharmacological interventions are employed that strongly support the reports that different RyR isoforms are involved in mediating nuclear vs subplasmalemmal Ca²⁺ release events and that a replete SR is required for generation of Ca²⁺ changes.

We thank this Reviewer for their kind comments above.

We would also like to take this opportunity to emphasise that the nanocourses we describe are not insulated from the “rest of the cytoplasm”, but do in fact demarcate much of the cytoplasm. We propose that in doing so each cytoplasmic nanocourse is insulated from the others under resting conditions. Organelles sit within them and migrate through them (see below).

While interesting, I don't consider that there is enough direct evidence to support the conclusions drawn.

*1. Cells are densely packed with organelles that limit diffusion of Ca²⁺. But despite this, Ca²⁺ changes are reported to be global in **nature**. These organelles are also constantly moving - this is well established for the ER and for mitochondria. As such the shape of the regions of the cytosol devoid of these organelles is also changing. These shape changes would also impact on the local concentration of the Ca²⁺ indicator dye employed. Together, these processes could therefore result in similar observations as described in this MS.*

The Reviewer's comment is understandable. However, it should be noted that unitary calcium signals at rest are local, not global. The only signals that propagate are those induced by Angiotensin II within the extra/perinuclear nanocourses (new Figure 5). All other calcium signals reported appear to be restricted to the nanocourses in which they arise (whether basal, maurocalcine-induced, or Angiotensin II-induced calcium signals within nuclear nanocourses).

To address this Reviewer's concern regarding organelles, we have now carried out additional experiments which show that the major organelles (endo/lysosomes and mitochondria) are positioned within the nanocourses. Furthermore:

(a) Small singular endo/lysosomes move freely through the nanocourses, while others form larger, more static clusters.

- (b) Mitochondria form static clusters in these differentiated cells, as described previously by others (Chalmers, S., Saunter, C., Wilson, C., Coats, P., Girkin, J. M., & McCarron, J. G. (2012). Mitochondrial motility and vascular smooth muscle proliferation. *Arterioscler Thromb Vasc Biol*, 32(12), 3000-3011. doi:10.1161/ATVBAHA.112.255174). They only migrate during proliferation.
- (c) The movement/activity of the relatively small number of motile endolysosomes might have highly localized effects, but in general this will likely impact calcium flux within a smaller proportion of cytoplasmic nanocourses and along only a proportion of their length. Furthermore the endo/lysosomes move at a different rate (≤ 0.5 microns in 4 seconds) than the frequency with which “hotspots” of calcium release fluctuate (0.16Hz).

See page 6, paragraph 1, lines 7-11, which now reads:

“Supporting the view that they represent a circuit for cell-wide communication, LysoTracker Red labelled endolysosomes migrated through this network of cytoplasmic nanocourses (Fig. 1I; 0.25 Hz sampling frequency for dual labelling; Supplementary movie 4). By contrast, in these differentiated cells MitoTracker Red labelled mitochondria formed static clusters, as reported previously by others ²⁶, that sat within the nanocourse network (Fig. 1J; Supplementary movie 5).”

2. Given the role of the cytoskeleton in mediating the movement of the ER (actin) and mitochondria (microtubules), it would be of interest therefore to prevent movement of these organelles.

As mentioned above, only a relatively small subset of endolysosomes are motile, the majority appear static in differentiated cells, over the time course of our experiments (Supplementary video 5). Moreover, all mitochondria remain static (Supplementary video 6).

In short, the majority of these organelles are static in these differentiated cells. Further to this, we would like to point out that wortmanin was found to disrupt nanocourses, while ML9 (myosin light chain kinase inhibitor) failed to block myocyte contraction at room temperature.

3. The MS reports hotspots of Ca²⁺ that are presumably generated from clusters of RyRs. these events should be analogous to Ca²⁺ sparks. Given that Ca²⁺ sparks have a defined lifetime and dimensions, more efforts should be made to capture these events using high speed imaging approaches. The direct measurement of Ca²⁺ sparks would strengthen the proposal that hotspots are bona fide Ca²⁺ release events.

Imaging at higher temporal resolution is absolutely required to determine whether these release events propagate and truly act as a signalling pathway.

As we report, it is MORE than likely that “hotspots” reflect “calcium sparks”. The duration, size and amplitude of hotspots are very similar to calcium sparks.

Given that all such measures previously reported using line scan confocal imaging are within the sampling frequency of our experiments (0.5 Hz), the measurements we have made are as good an approximation as previously published data by line scan. Therefore, it is not necessary to complete line scan as it would offer little over and above previously published data from the same and different cells. Moreover, it would be very difficult to complete line and z scan within time windows close enough to deliver any significant advantage. See page 5, paragraph 1, line 7 to 20, which now reads:

“During short time series’ (2-6min; *note*, experiment duration limited by photo-toxicity) hotspots of local Ca^{2+} flux, $\approx 200\text{-}400$ nm in diameter, were readily identifiable in pseudocolour representations of this cell-wide network at rest (Fig. 1E), and did not appear to propagate beyond the nanocourse within which they arose. Each individual hotspot of Ca^{2+} flux exhibited asynchronous temporal characteristics when compared to adjacent hotspots within the same nanocourse, or hotspots arising in different nanocourses (Fig. 1E-F; Supplementary Video 2-4). Such activity was not evident in averages of Fluo-4 fluorescence for any given nanocourse as a whole (Fig. 1F, lower panels). The fluorescence intensity of nanocourse hotspots fluctuated ($\Delta F_x/F_0$: peripheral 0.18 ± 0.02 ; extraperinuclear 0.15 ± 0.01 ; perinuclear 0.16 ± 0.02 ; nuclear 0.15 ± 0.02) with a frequency of $\approx 0.16\pm 0.01$ Hz irrespective of the region of the cell in which they were recorded (sampling frequency = 0.5 Hz). Both asynchronicity at rest and spatiotemporal characteristics similar to those of these “hotspots” have been observed by others using line-scan imaging of spontaneous “ Ca^{2+} sparks” arising at RyR clusters in arterial smooth muscles and other cell types: half maximum full width ≈ 1 μm , frequency $\approx 0.2\text{-}0.5$ Hz and amplitude ($\Delta F_x/F_0$) ≈ 0.8 ^{3, 26, 27}”

4. Fluo 4 signals are not calibrated and response to Ca is known to be affected by concentration and local environment. Does the Ca^{2+} indicator report Ca^{2+} similarly in all types of hotspots. It is well established that the environment of the nucleus affects the Ca^{2+} dependent changes in fluo 4 - for e.g. basal fluorescence is higher than in the cytosol. Could the authors assess the dynamic range and Ca^{2+} sensitivity of their indicator in the hotspots analysed.

As the Reviewer states, it is well established that the local environment will alter basal fluorescence. Whatever the reason, this has allowed us to distinguish nuclear invaginations from the nucleoplasm and from cytoplasmic nanocourses that sit outside the nucleus, because the basal level of fluorescence is consistently higher in nuclear nanocourses (invaginations). This is helpful, rather than being a hinderance. See page 4, paragraph 2, lines 16-20, which now read:

“This suggested that invaginations of the nuclear envelope might demarcate discrete signalling compartments that could be observed without the need for further image processing, irrespective of whether or not differences in fluorescence intensity resulted from differences in local cytoplasmic Ca^{2+} concentration or the influence of the local environment within each of these compartments on general Fluo-4 fluorescence characteristics ^{20, 21}.”

In cultured, proliferating smooth muscle cells by contrast, nuclear Fluo-4 fluorescence is higher than cytoplasmic Fluo-4 fluorescence, as we show in the present paper (see Figure 6). However, the basal fluorescence of Fluo-4 within the nucleoplasm of differentiated smooth muscle cells is lower than the “cytoplasmic nanocourses” and “nuclear invaginations”, as we report (see Figures 1-5; Supplementary Figure 3-5, 8 & 9).

Calibration of calcium concentration within nanocourses is somewhat futile. The local volume is likely in the picolitre range (a rough calculation based on dimensions from either EM or live cell imaging tells us this). A further rough calculation tells us that addition of 1 or 2 calcium ions to a 1 picolitre volume will raise the “concentration” from nanomolar to micromolar. We have estimated this previously and cite the paper in the relevant part of our discussion (Fameli, Ogunbayo, van Breemen, & Evans, 2014). As we propose, this likely allows points of calcium flux (hotspots) to act as highly localised “switches”.

Fameli, N., Ogunbayo, O. A., van Breemen, C., & Evans, A. M. (2014). Cytoplasmic nanojunctions between lysosomes and sarcoplasmic reticulum are required for specific calcium signaling. *F1000Res*, 3, 93. doi:10.12688/f1000research.3720.1

See page 12, paragraph 2, line 1 onwards, which now reads:

“Regardless of the functional subdivision of nanocourses, all path lengths from Ca^{2+} release site to targeted signalling complexes must be on the nanoscale, with picolitre volumes of cytoplasm lying within the boundaries of each nanocourse⁵². Relatively small net increases in local Ca^{2+} flux (1-2 ions per picolitre) will therefore be sufficient to raise the local concentration into the affinity ranges of most cytoplasmic Ca^{2+} binding proteins⁵². Ca^{2+} binding proteins may thus operate as local “switches” that coordinate nanocourse-specific functions, the probability of moving from *OFF* to *ON* determined by changes in unitary rather than macroscopic Ca^{2+} flux.”

5. To compensate for local concentration differences of the dye, an approach where a second non Ca sensitive dye is co-loaded may help.

I am not sure what could be gained from this, given the above (see point 4). Nevertheless, we have now completed experiments with co-labelling of mitochondria, lysosomes and ER membranes. Cell-wide nanocourse networks are very much in evidence in all cases, and the nuclear nanocourses always exhibit higher basal fluorescence than the nucleoplasm, in differentiated cells. See new Figure 1 and new supplementary Figure 2, 8 and 9.

6. It is not clear to me at present whether these hotspots are stochastic release events with no real function or have a true function.

The hotspots of calcium flux likely maintain the “resting state” of the cell under the conditions of our experiments. Supporting this, we show very clearly that re-directing calcium flux in different ways using different stimuli leads to relaxation (Figure 2, Maurocalcine) or contraction (Figure 5, Angiotensin II). In

each case the induced cellular responses are blocked by tetracaine and thapsigargin, demonstrating that calcium flux from the SR through RyRs underpins each response. The same agents (tetracaine and thapsigargin) attenuate or block hotspots at rest (although here the pre-incubation times mitigate against analysing the impact of the blocking agent on cell shape and size).

We also now show that tetracaine block of calcium flux represses S1009A expression (see new Fig. 4), as does loss of nuclear invaginations on proliferation of these cells in culture (see new Fig. 6), i.e. invaginations and calcium flux across their outer nuclear membrane impacts gene expression (see point 7 below).

7. The authors provide correlation of the nanocourse with location of the epigenetic marks H3K9me2 and H3K9me3, which are known to be enriched under the nuclear envelope and mediate gene repression. Genes covered by H3K9me3 are particularly associated with heterochromatin and permanently silenced in differentiated cell types. While the observations are interesting, there is no link between Ca²⁺ change and dynamic alteration of the abundance of these marks or activation of gene expression.

We now provide measures of expression changes by qRT-PCR and RNAscope, which confirm that expression of genes repressed by H3K9me2 / Baf is indeed changed by altering calcium flux through RyRs, because blocking these channels with tetracaine promotes their repression. See page 8 paragraph 3, through page 9, paragraph 1, which reads:

“The functional reasons for the isolation of nuclear nanocourses are not clear, but it may be to prevent wide-scale gene activation / inactivation events that could switch cells from a differentiated to proliferative phenotype, operated through specific changes in Ca²⁺ flux. Normal ovoid nuclei tend to have histones carrying both H3K9me2 and H3K9me3 marks, and the chromatin cross-linking protein barrier to auto-integration factor (BAF) associating with NE proteins such as emerin and making the nuclear periphery generally silencing^{33, 34, 35}. However, interestingly, these marks segregate in differentiated arterial myocytes with the H3K9me2/3 both still at the outer limits of the nucleus but depleted with respect to BAF, and the nuclear invaginations rich with H3K9me2 (Fig. 4C-E) and BAF (Fig. 4F-H) but depleted with respect to H3K9me3 (Supplementary Fig. 6). The combination of H3K9me2/3 together is strongly silencing, but absent the me3 mark and the me2 can reflect a poised state that has been found at myogenic regulators such as the myogenin promoter³⁶. It is possible that the non-propagating Ca²⁺ transients in distinct invaginations in some way specifically regulate chromatin in differentiated cells as the different chromatin marks are concentrated in puncta (Fig. 4C-H): discrete H3K9me2-lamin A puncta (471±38 nm in diameter) were separated by 335±46 nm, while emerin-BAF puncta (361±41 nm in diameter) were separated by 495±61 nm, approximating the 350 nm spacing between RyR1 clusters (tetracaine-sensitive hotspots) of nuclear nanocourses. Potentially Ca²⁺ responsive and functionally distinct chromatin domains may therefore be established by nuclear invaginations. Supporting this, qRT-PCR (Fig. 4I) and RNAscope® (Fig. 4J) showed that blocking Ca²⁺ flux through RyRs with tetracaine (1mM, 90min pre-incubation) reduced the expression of two genes of interest (identified by

RNAseq), one encoding the DNA mismatch repair protein MutL homolog 1 (*Mlh1*), which can be repressed through interaction with H3K9me2^{37, 38}, and another encoding the S100 calcium binding protein A9 (*S100a9*), which can be repressed by BAF³⁹. That said, further investigation into the role of nuclear invaginations in regulating gene expression will undoubtedly reveal greater complexities of gene regulation, given that individual, acutely isolated smooth muscles possess different types (Supplementary Fig. 7) and different numbers of lamin A and emerin positive invaginations.”

9. The authors propose that the effects of AngII are mediated via cADPR and RyRs. while cADPR can directly activate RyR1, it only acts to sensitise RyR2. Based on this differential sensitivity, the authors propose the differential roles of RyR1 and 2 in mediating Ca changes in different cellular compartments.

RyRs open with a defined open probability - they are not permanently closed but undergo stochastic openings in a manner that is increased by Ca. while the authors suggest that the reason why RyR2 are not activated by AngII is because cADPR does not directly activate them. However, if these channels are undergoing spontaneous release events, wouldn't cADPR not act to enhance channel coupling subsequent to a stochastic opening thereby generating a hotspot?

The Reviewer is quite correct in noting that RyRs open with a defined open probability. The situation we see here is no different. As we state, “hotspots” of calcium flux rise and fall with spatiotemporal characteristics consistent with those of calcium sparks. We clearly state that this can be seen in all nanocourses, irrespective of whether RyR1, RyR2 or RyR3 is the predominant subtype by region.

We do not suggest that RyR2 are not activated by AngII, we propose the opposite. That RyR2s are activated by AngII, albeit indirectly, and that it is RyR2 that carries the propagating wave induced by AngII. We apologise for any confusion caused by any lack of clarity on this point. We have revised this section extensively to improve delivery of our message.

What we had also argued was that RyR2 activation was dependent in some way on cADPR sensitizing RyR2 to activation by CICR, with the trigger calcium being derived from another source. What we had failed to do was elaborate on the mechanism by which cADPR elicits a propagating Ca²⁺ wave. We thank the Reviewer for highlighting this fact!

We have previously established that cADPR elicits quite different Ca²⁺ signals when applied by intracellular dialysis into pulmonary arterial myocytes at different concentrations.

Intracellular dialysis of high concentrations (100 micromolar) of cADPR triggers a global calcium wave in pulmonary arterial smooth muscle cells, through RyR activation (Ogunbayo et al., 2018; Wilson et al. 2001). Consistent with this, we now show that 8-bromo-cADPR blocks propagating calcium waves induced by AngII.

Accordingly, we have previously shown that 8-bromo-cADPR blocks hypoxic pulmonary vasoconstriction (Dipp and Evans, 2001; Wilson et al., 2001).

In complete contrast, intracellular dialysis of low concentrations (10 micromolar) of cADPR preferentially mobilises calcium from the SR proximal to

the plasma membrane through RyR1s, thus inducing membrane hyperpolarization and vasodilation (Boittin et al., 2003; Clark et al., 2010). We have previously shown that RyR1s and RyR3s can be activated by cADPR (Boittin et al., 2003; Clark et al., 2010; Ogunbayo et al., 2011; all now cited), while others have demonstrated that cADPR sensitises RyR2 to CICR (Cui et al., 1999). We have revised our discussion accordingly. See page 9, last paragraph, lines 13 to 20, through Page 10, first paragraph, lines 1 to 10.

“This is significant given that RyR2 and RyR3 are preferentially targeted to extraperinuclear and perinuclear regions, respectively, while RyR1 clusters predominate in subplasmalemmal regions and nuclear invaginations^{16, 18}, because it is RyR2 and RyR3, but not RyR1, that hold the capacity to carry propagating waves by Ca²⁺-induced Ca²⁺ release (CICR)^{17, 29, 44, 45}. Irrespective of cellular region, increases in Ca²⁺ flux induced by Angiotensin II were abolished by prior depletion of SR stores with thapsigargin, block of RyRs with tetracaine and by pre-incubation with the cyclic ADP-ribose antagonist 8-bromo-cADPR (Fig. 5D-E), which is in line with the fact that 8-bromo-cADPR also blocks hypoxic pulmonary vasoconstriction at the level of the smooth muscle^{46, 47}. Accordingly, we have previously shown that intracellular dialysis of high concentrations of cADPR (100 μM) evokes a global Ca²⁺ wave and contraction of acutely isolated pulmonary arterial myocytes^{47, 48}. Given that cADPR preferentially activates RyR1s and RyR3s⁴⁹ but can only sensitise RyR2s to CICR⁴⁴, it therefore seems likely that cADPR generated within or proximal to extraperinuclear nanocourses in response to Angiotensin II, serves to activate local subpopulations of RyR1s and/or RyR3s⁴⁹ while delivering concomitant sensitisation of RyR2s to CICR⁴⁴ that permits subsequent initiation of a propagating Ca²⁺ signal and thus myocyte contraction. This is in stark contrast to the effect of low concentrations of cADPR (10 μM), the intracellular dialysis of which preferentially releases Ca²⁺ from RyR1s on the peripheral SR proximal to the plasma membrane, to thus evoke membrane hyperpolarization and vasodilation^{15, 16}.”

10. *This effect of cADPR could also be affected by the density of RyR expression. Recent evidence from cardiac myocytes - modelling and imaging - has shown a key role for the size of the cluster and its organisation in generating Ca²⁺ sparks. The more organised the cluster, the more efficient it is in generating sparks. As such, larger clusters with more spontaneous openings could be more sensitive to cADPR than smaller disorganised clusters. Could the authors therefore examine RyR clusters dimensions in their cells.*

As mentioned above, it would appear that the determining factor for the induction of cADPR-dependent, propagating calcium waves is the local concentration of cADPR, rather than RyR cluster size per se.

11. *It would be useful to probe the effect of Ca²⁺ buffering with BAPTA on the proposed differential activation of RyR1 and 2. IF directly activated by cADPR, BAPTA should be without effect whereas activation of channels/hotspots by CICR would be reduced.*

As mentioned above, the determining factor for the induction of cADPR-dependent, propagating calcium waves is the local concentration of cADPR. Moreover, previous studies have demonstrated that BAPTA blocks the effect of intracellular dialysis of cADPR (Boittin et al 2003). BAPTA is a fast chelator of calcium and would be expected to have equivalent effects to teracaine, in the presence of which we are unable to reliably identify cytoplasmic nanocourses or hotspots within peripheral (subplasmalemmal), extraperinuclear, or perinuclear nanocourses.

12. A similar effect of cADPR could also be brought about by IP3 induced Ca release generated downstream of AngII. This effect of IP3 would not need to be great to sensitise RyRs and bring about a Ca²⁺ change. While the authors may have probed the role of IP3 in other work and in nuclear Ca regulation in this MS, the authors should also test the role of IP3 in the Ca²⁺ changes in the nanocourses reported in this MS.

We examined this, but found no effect of 2-APB. We do not rule out a role for IP₃Rs, which are present in these cells. However, they appear to be in a “different space”, spatially segregated from and unable to couple to RyRs, as we have reported previously (Boittin et al 2002). This is now discussed on page 10, paragraph 1, lines 10-14.

“Furthermore, the response to Angiotensin II remained unaffected in the presence of the inositol (1,4,5) trisphosphate receptor (IP₃R) antagonist 2APB (not shown). This is in accordance with the fact that IP₃Rs do not couple by CICR to RyRs in pulmonary arterial myocytes⁴⁴, and suggests that this segregation of RyRs from IP₃Rs might be conferred by the targeting of RyR2/RyR3s to the SR that demarcates cytoplasmic nanocourses.”

At this time, we can provide no definitive explanation for this finding other than that given above, but now refer to this previous work on page 10, lines 10-14:

Boittin FX, Galione A, Evans AM. Nicotinic acid adenine dinucleotide phosphate mediates ca²⁺ signals and contraction in arterial smooth muscle via a two-pool mechanism. *Circ Res.* 2002;91:1168-1175

13. The authors indicate that the hotspots asynchronously activate. What is the reason for this? As Ca²⁺ can promote its own release, I would think it likely that the nanocourses should also facilitate coupling between RyR clusters (that underlie the hotspots). The temporal resolution of imaging is clearly substantially too slow to detect all release events if these hotspots are arising from a cluster of RyRs in a manner analogous to sparks.

The temporal resolution of our studies (0.5Hz) is sufficient to record the lifetime of calcium sparks. As we report in this paper it is likely that “hotspots” = “calcium sparks”. Given this fact our, demonstration of asynchronous “hotspot” activation is entirely consistent with previously published studies on RyRs, irrespective of subtype. These show a predominance of asynchronous activation of adjacent sparks (RyR clusters) at rest, with few instances of

increased probability of opening, but no spontaneous signal propagation for cardiac RyR2s.

Furthermore, it is evident that nanocourses have the capacity to restrict wider Ca^{2+} diffusion.

See page 5, paragraph 1, line 10 to 19, which now reads:

“Each individual hotspot of Ca^{2+} flux exhibited asynchronous temporal characteristics when compared to adjacent hotspots within the same nanocourse, or hotspots arising in different nanocourses (Fig. 1E-F; Supplementary Video 2-4). Such activity was not evident in averages of Fluo-4 fluorescence for any given nanocourse as a whole (Fig. 1F, lower panels). The fluorescence intensity of nanocourse hotspots fluctuated ($\Delta F_x/F_0$: peripheral 0.18 ± 0.02 ; extraperinuclear 0.15 ± 0.01 ; perinuclear 0.16 ± 0.02 ; nuclear 0.15 ± 0.02) with a frequency of $\approx 0.16 \pm 0.01$ Hz irrespective of the region of the cell in which they were recorded (sampling frequency = 0.5 Hz). Both asynchronicity at rest and similar spatiotemporal characteristics have been observed by others using line-scan imaging of spontaneous “ Ca^{2+} sparks” arising at RyR clusters in arterial smooth muscles and other cell types: half maximum full width $\approx 1 \mu\text{m}$, frequency ≈ 0.2 - 0.5 Hz and amplitude ($\Delta F_x/F_0$) ≈ 0.8 ^{3, 26, 27}.”

14. Can the authors also rule out the possibility that the Ca^{2+} indicators used are also not acting to restrict diffusion of the Ca^{2+} release events. As suggested above, the concentration of the indicator may be substantially higher in the nanocourses than what is usually observed in the bulk cytosol.

All calcium indicators will restrict diffusion to some extent. However, it is clear that signals can propagate when induced by stimuli that are known to elicit propagating signals, such as AngII. This stimulus elicits calcium signals of equal if not greater magnitude in nuclear nanocourses than it does in extra/perinuclear nanocourses, but calcium signals arising in nuclear nanocourses do not propagate beyond the nanocourses in which they arise. In short, in the absence of a determining stimulus, signal propagation is restricted, but not by the indicator.

14. These observations have all been made in smooth muscle cells and it is interesting that effects are lost on culture. Could the authors provide information whether the nanocourses are a smooth muscle specific phenomenon or are they of wider importance. If not could the authors explicitly state that the nanocourses are a particular feature of smooth muscle cells.

We have now explicitly stated that the observations made are a specific feature of pulmonary arterial smooth muscles, because different smooth muscles have different arrangements of intracellular calcium release channels, as do other cell types.

That said, nuclear invaginations are certainly widespread, being observed in many different cell types, and this fact is stated clearly in the introduction, results and discussion. Junctions between the plasma membrane and the S/ER are also evident in many different cell types. Therefore, we have indicated that some if not all of our findings are of general significance. Supporting this, we

cite a number of relevant papers that cover neurons, cardiac, skeletal and smooth muscles.

We have extended our discussion on the nature and distribution of S/ER junctions within other cell types to highlight that junctional organisation likely varies with cell type / function, i.e. there will likely be key cell-specific differences; as noted by this Reviewer, our studies on proliferating cells in culture suggest this. Detailed additional investigations will be required to define networks relevant to other cell types, sufficient to warrant stand-alone publications.

We have revised our concluding paragraph to reflect this, page 13, first paragraph, last 5 lines, which now reads:

“Perhaps more importantly, these network activities are not hardwired, reconfiguring to deliver different outputs during phenotypic modulation on the path, for example, to cell proliferation. This in itself suggests that cytoplasmic nanocourses may be common to but vary in nature between different cell types. Supporting this, nuclear envelope invaginations are a feature of many cell types^{9, 10, 11, 12, 13} while other junctional complexes of the S/ER vary by cell type and even between different smooth muscles^{2, 22}.”

15. The quality of image capture is key to this study. In the absence of sufficient image resolution or over iteration in deconvolution, regions of fluorescence can be blurred to an extent that they merge. Could the authors therefore provide further information regarding their parameters of image capture - specifically, pixel size in x, y and z. this is required for the Ca imaging - I presume with the Nikon A1R and also for the immunofluorescence analysis using the Deltavision system.

We now provide these details in the methods section (see confocal calcium imaging).

17. For the Ca²⁺ imaging was a z stack collected and then deconvolved to give a slice or is deconvolution in 2D only?

2D deconvolution was utilised. 3D z stacks of ER tracker in particular are compromised by photobleaching of reporter dyes, in a manner exacerbated by increasing sample number.

18. Given the z resolution of the system is likely closer to 600 nm, the authors cannot rule out Ca hotspots, fluorescence changes arising from under the plane of focus.

We cannot rule out the possibility that out of focus light will be gathered to some degree, any more than others who use confocal imaging to obtain line scan images of calcium sparks (most of these previously published studies using line scan having been completed without deconvolution).

However, the approaches used here will have insured that out of focus light has been minimised. That hotspots identified arise within the volume sampled is even more likely given that they can be distinguished from the remaining length of the nanocourse in which they arise, and exhibit temporal characteristics consistent with calcium sparks. That this is the case is confirmed by the fact

that signal averages for whole nanocourses do not exhibit temporal fluctuations in Fluo-4 fluorescence observed for hotspots (see Fig. 1).

19. Could the authors clearly indicate which images are captured with which system in the figure legends.

This has now been clearly indicated in the legend for Figure 1.

Minor:

1. Page 1 of the introduction the authors state: 'Evidence suggests that within the sarcoplasmic reticulum (SR), the primary intracellular Ca²⁺ store, multiple releasable pools of Ca²⁺ exist'.

Is this really true? Or is it that Ca²⁺ can be released by a number of different mechanisms. In other cell systems, it has been shown that the ER is a contiguous organelle.

We thank the Reviewer for pointing out that the sentence construction here could be misleading. The SR is indeed a contiguous organelle, within which the spatial organisation of different SERCA types supports SR refilling at different regions of the cell and thus the release of calcium from these different regions, in some cases through different channels types. These different release sites could be considered to be operating as “independent pools” given the likely impact of calcium binding proteins and tortuosity on calcium movements between different regions of the SR. That said, we have revised our introduction, see page 3 paragraph 2 and 3:

“The primary intracellular Ca²⁺ store is the sarco/endoplasmic reticulum (S/ER)², which is known to be a contiguous organelle, from its origin at the outer nuclear membrane to the periphery of the cell. Yet the S/ER delivers Ca²⁺ signals with clear diversities of form and function^{4, 5}. In arterial smooth muscles, for example, the current consensus is that relaxation is mediated by highly localized Ca²⁺ sparks that recruit Ca²⁺-activated potassium channels to promote plasma membrane hyperpolarization, while contraction is triggered by propagating global Ca²⁺ waves⁸, with adjustments to gene expression presumed to be governed by the spatiotemporal patterns of global Ca²⁺ transients that gain unrestricted entry to the nucleoplasm across the nuclear envelope and its invaginations^{9, 10, 11, 12, 13}.

However, in smooth muscles it has long been suggested that multiple, spatially segregated and independently releasable subcompartments of Ca²⁺ may exist within the SR, filled by spatially segregated subtypes of sarco/endoplasmic reticulum Ca²⁺ ATPase (SERCA) pumps and mobilised through similarly segregated subtypes of Ca²⁺ release channel^{14, 15, 16}, including ryanodine receptors (RyRs) 1, 2 and 3^{17, 18, 19, 20, 21}. This led to an alternative proposal, that different Ca²⁺ signals may arise in distinct cytoplasmic spaces demarcated by junctions between the SR and its target organelles^{22, 23}. However, direct visualization of Ca²⁺ signalling within junctional complexes of the SR has yet to be achieved, so little more than speculation has guided such considerations on functional signal segregation within cells²².”

2. Page 4. A space should always be present between unit and integer.

Noted and corrected.

3. Different SERCA isoforms are reported in different subcellular compartments. What would be the reason for this - what could be the benefit? As different SERCA isoforms have different Ca affinities and transport rates, different rates of recovery of the Ca²⁺ signals may be observed at different cellular locations. Could the authors please examine.

The reviewer is correct in this respect. We have previously published work on this question and have written a number of review articles that discuss this matter. The relevant papers are cited here and discussed:

- (a) Boittin, F. X., Dipp, M., Kinnear, N. P., Galione, A. & Evans, A. M. Vasodilation by the calcium-mobilizing messenger cyclic ADP-ribose. *The Journal of biological chemistry* **278**, 9602-9608 (2003).
- (b) Clark, J. H. *et al.* Identification of Functionally Segregated Sarcoplasmic Reticulum Calcium Stores in Pulmonary Arterial Smooth Muscle. *The Journal of biological chemistry* **285**, 13542-13549 (2010).
- (c) Evans, A. M. Nanojunctions of the Sarcoplasmic Reticulum Deliver Site- and Function-Specific Calcium Signaling in Vascular Smooth Muscles. *Adv Pharmacol* **78**, 1-47, doi:10.1016/bs.apha.2016.10.001 (2017).

That the temporal characteristics might be affected by the SERCA pump present in a given location is evident from the fact that even when we analyse calcium signals in nuclear invaginations (SERCA1) of cells without deconvolution, the rate of decay is different from that observed in the wider cell (SERCA2a/b).

“Multiple coordinated actions may thus be delivered by signal segregation between distinct nanocourse networks, enabling nanocourse-specific delivery of Ca²⁺ signals with distinct temporal characteristics (Fig. 4 and 5, Supplementary Fig. 9).”

Reviewer #2 (Remarks to the Author):

Duan et al. extensively examined arterial smooth muscle cells using various techniques including state-of-the-art approaches, and report the distinct residency and functional role of RyR subtypes. Their observations are definitely novel and likely make a big deal, which would fundamentally elevate our understanding of cellular Ca²⁺ signaling in smooth muscle.

We thank this Reviewer for these kind and supportive comments.

However, several improvements may be still needed.

1. Are the home-made antibodies against RyRs sure in subtype specificity? This point must be very important because the results largely depend on the subtype-specific antibodies. For example, can the authors present WB and IHC data in skeletal and heart muscles? It has been already established that skeletal muscle contains

abundant RyR1 and also a small amount of RyR3, while cardiac muscle specifically contains RyR2.

The “home-made antibodies” used here were provided by Sidney Fleischer. These are sequence-specific, affinity purified antibodies. They are recognised in the field as being highly specific and the “gold standard”. No commercially available antibodies have this level of specificity for rodent RyRs by subtype. The papers in which these antibodies have been characterised and used previously are all cited at the appropriate points in our paper.

- (d) Jeyakumar, L. H. *et al.* The skeletal muscle ryanodine receptor isoform 1 is found at the intercalated discs in human and mouse hearts. *J Muscle Res Cell Motil* **23**, 285-292 (2002).
- (e) Jeyakumar, L. H. *et al.* FKBP binding characteristics of cardiac microsomes from diverse vertebrates. *Biochemical and biophysical research communications* **281**, 979-986, doi:10.1006/bbrc.2001.4444 (2001).
- (f) Jeyakumar, L. H. *et al.* Purification and characterization of ryanodine receptor 3 from mammalian tissue. *The Journal of biological chemistry* **273**, 16011-16020 (1998).
- (g) Clark, J. H. *et al.* Identification of Functionally Segregated Sarcoplasmic Reticulum Calcium Stores in Pulmonary Arterial Smooth Muscle. *The Journal of biological chemistry* **285**, 13542-13549 (2010).

2. Both RyR2 and RyR3 have been frequently examined in smooth muscle, however, RyR1 was not. Is it sure that arterial muscle contains all RyR subtypes? Which subtype expression is altered during the proliferating stage? To re-confirm such points, RT-PCR experiments using subtype-specific primers are probably useful.

RyR1, RyR2 and RyR3 have previously been shown to be present in a variety of smooth muscles including pulmonary arterial myocytes, by ourselves and others. In some cases using RT-PCR and western blot. See for example:

- A. Clark, J. H. *et al.* Identification of Functionally Segregated Sarcoplasmic Reticulum Calcium Stores in Pulmonary Arterial Smooth Muscle. *The Journal of biological chemistry* **285**, 13542-13549 (2010).
- B. Lifshitz, L. M., Carmichael, J. D., Lai, F. A., Sorrentino, V., Bellve, K., Fogarty, K. E., & ZhuGe, R. (2011). Spatial organization of RYRs and BK channels underlying the activation of STOCs by Ca(2+) sparks in airway myocytes. *J Gen Physiol*, *138*(2), 195-209. doi: 10.1085/jgp.201110626
- C. Herrmann-Frank, A., Darling, E., & Meissner, G. (1991). Functional characterization of the Ca(2+)-gated Ca2+ release channel of vascular smooth muscle sarcoplasmic reticulum. *Pflugers Arch*, *418*(4), 353-359.
- D. Gilbert, G., Ducret, T., Marthan, R., Savineau, J. P., & Quignard, J. F. (2014). Stretch-induced Ca2+ signalling in vascular smooth muscle cells depends on Ca2+ store segregation. *Cardiovasc Res*, *103*(2), 313-323. doi: 10.1093/cvr/cvu069
- E. Yang XR, *et al.* Multiple ryanodine receptor subtypes and heterogeneous ryanodine receptor-gated Ca2+ stores in pulmonary arterial smooth muscle cells. *Am J Physiol Lung Cell Mol Physiol* **289**, L338-348 (2005).

We have now revised the introduction to make this clear, see page 3, paragraph 3, which now reads:

“However, in smooth muscles it has long been suggested that multiple, spatially segregated and independently releasable subcompartments of Ca^{2+} may exist within the SR, filled by spatially segregated subtypes of sarco/endoplasmic reticulum Ca^{2+} ATPase (SERCA) pumps and mobilised through similarly segregated subtypes of Ca^{2+} release channel^{14, 15, 16}, including ryanodine receptors (RyRs) 1, 2 and 3^{17, 18, 19, 20, 21}. This led to an alternative proposal, that different Ca^{2+} signals may arise in distinct cytoplasmic spaces demarcated by junctions between the SR and its target organelles^{22, 23}. However, direct visualization of Ca^{2+} signalling within junctional complexes of the SR has yet to be achieved, so little more than speculation has guided such considerations on functional signal segregation within cells²².”

3. After publication of this study, without reservation, many researchers in the Ca^{2+} signaling field will focus on the molecular mechanisms underlying the RyR subtype-specific characteristics, such as subcellular localization and functional channel crosstalk in smooth muscle. The authors can propose and explain such mechanisms in the Discussion part?

We do touch on this in the discussion and present a schematic depicting the overall mechanisms. However, for the sake of brevity we refer the reader to our more detailed discussion of these matters in a recent review article. See for example:

Evans, A. M. Nanojunctions of the Sarcoplasmic Reticulum Deliver Site- and Function-Specific Calcium Signaling in Vascular Smooth Muscles. *Adv Pharmacol* **78**, 1-47, doi:10.1016/bs.apha.2016.10.001 (2017).

For the purposes of this manuscript we submit a minimal model that describes the proposed organisation of RyR and SERCA within the junctional complexes described here (see supplementary Figure 12C)

4. By using commercial microscopes, optical resolution is limited and deconvolution technique has been recently developed. Deconvolution seems to work well in previously reported cases, while inappropriate deconvolution should mess up observations. Therefore, the procedure for deconvolution is critical to reproduce the results and to convince general readers. The basic procedure of the deconvolution utilized should be briefly described in the Method section.

The deconvolution used here is now described in the methods (see for example, confocal calcium imaging).

5. “Quantum tunnelling” is a provocative term, but it seems to be a defined technical term in the quantum mechanics of physics. If the authors simply use it as paraphrase of “ Ca^{2+} tunnelling”, “quantum” cannot be appropriate here. In addition, “calcium tunnels” has been previously proposed by Petersen et al. (*Cell Calcium* **42**, 373-378, 2007). To distinguish the reported concept from the Petersen’s proposal, the authors may need other words in this study.

We understand this Reviewer’s point, and now make it clear that the process we have observed is “analogous to quantum tunnelling”. We now refer to the signalling mechanism we observe as “modulated quantum calcium flux”. See page 12, paragraph 2, last 10 lines, which now reads:

“This draws obvious parallels (Supplementary Fig. 13) to mechanisms of conduction in single-walled carbon nanotubes, which behave as quantum wires that transmit charge carriers through discrete conduction channels, enabling memory, logic and parallel processing. Thus, by analogy, our observations point to the incredible signalling potential that may be afforded by modulating “quantum Ca²⁺ flux” on the nanoscale, in support of network activities within cells with the capacity to permit stimulus-dependent orchestration of the full panoply of diverse cellular processes. Perhaps more importantly, these network activities are not hardwired, reconfiguring to deliver different outputs during phenotypic modulation on the path, for example, to cell proliferation.”

Reviewers' comments:

Reviewer #1 (Remarks to the Author):

The authors have made a substantial effort to address issues raised by myself and the second referee. This is much appreciated. It is also clear the authors have taken the comments on board and made a serious attempt to address my concerns.

The visualisation of motile vs non motile organelles is interesting as is the direct link to gene expression. This latter observation truly presents the possible physiological importance of the observation reported.

I do however have a significant remaining concern that the authors did not adequately address. In particular whether the hotspots observed are truly calcium signals. The pharmacology of course is consistent with their existence but in the absence of true measurements of the time course of the events, it is not possible to say that they are or not.

The authors used an imaging frequency for analysis of the events of 0.5 Hz - 1 frame every 2 seconds. The integration period of these images is not shown.

A higher temporal resolution of imaging is required to capture the events.

The purpose of higher temporal image is not only for detection of an event but also to further establish that the event is a bona fide event. Without knowing the kinetics (or several time points on the event, it is not clear what these hotspots are - maybe regions where dye shows greater fluorescence or a movement artefact.

Sparks should have a time course that has a duration at half max of 40-50 ms. The authors indicate that the events are similar to sparks ('As we report, it is MORE than likely that "hotspots" reflect "calcium sparks". The duration, size and amplitude of hotspots are very similar to calcium sparks.') but this cannot be suggested without recording at a minimum of 100 Hz.

Line scanning is not the only option for image capture - although this does give good temporal resolution. The A1R confocal in resonant mode - band scan and averaging could give you some information at 100 Hz.

Reviewer #2 (Remarks to the Author):

All my critiques have been addressed. In response to several comments, the manuscript has been improved and now give an excellent demonstration of authors' conclusion.

Rebuttal to Reviewers Comments

Reviewer #1 (Remarks to the Author):

1. The authors have made a substantial effort to address issues raised by myself and the second referee. this is much appreciated. It is also clear the the authors have taken the comments on board and made a serious attempt to address my concerns.

We thank Reviewer 1 for recognising our efforts made.

2. The visualisation of motile vs non motile organelles is interesting as is the direct link to gene expression. This latter observation truly presents the possible physiological importance of the observation reported.

We thank Reviewer 1 for acknowledging this, it was tough work.

3. I do however have a significant remaining concern that the authors did not adequately address. In particular whether the hotspots observed are truly calcium signals. The pharmacology of course is consistent with their existence but in the absence of true measurements of the time course of the events, it is not possible to say that they are or not.

We now understand the depth of Reviewer 1's concerns. The plots of asynchronous Hotspot activity that we previously provided (Fig. 1F) are clearly insufficient for this, even to us.

4. The authors used an imaging frequency for analysis of the events of 0.5 Hz - 1 frame every 2 seconds. the integration period of these images is not shown.

I presume that Reviewer 1 is requesting the integration time at each pixel and the image size. The integration time for our Nikon A1R+ system is equal to the pixel time. It is stated in Nikon's system specifications that this, in part, allows for greater signal-to-noise ratios and spatial resolution than was available with previous confocal systems. We now provide this and other information within our methods. We trust that this meets with the Reviewers satisfaction.

Page 19, paragraph 2, last 4 lines:

“Images were acquired using the Galvano scanner (400Hz). Laser power was <3% (offset = 0), Hv \approx 120V, image size 1024x1024 (16 bit), frame time = 2s, sampling frequency 0.47Hz (1.1 frames s⁻¹). Note: greater spatial resolution is afforded by the Nikon A1R+ due, in part, to the fact that the integration time is equal to the pixel time (for further information see: https://m.nikoninstruments.com/content/download/15419/.../file/A1_2CE-SCNH-4r.pdf; <https://igmm-imaging.squarespace.com/s/Nikon-A1R-User-Manual-web.pdf>). Deconvolution was completed with pinhole at 0.7 Airy units, and pixel settings of 60 nm in x,y and 150 nm in z. After deconvolution under these conditions, (488 nm excitation, 2D) the estimated resolution is XY = 230 nm and Z = 582nm. We excluded from analysis any cell that exhibited contraction or other movement artefact sufficient to alter the focal plane and/or compromise our ability to take reliable measurements of regions of interest.”

5. A higher temporal resolution of imaging is required to capture the events.

Line scanning is not the only option for image capture - although this does give good temporal resolution. The A1R confocal in resonant mode - band scan and averaging could give you some information at 100 Hz.

The higher temporal resolution provided by increased sampling frequencies carries the cost of poorer signal-to-noise, and loss of the spatial resolution required to image either the “nanocourses” or the “hotspots” described here. We had actually begun our studies by using higher scan speeds afforded by the resonant scanner, as our confocal manager was extolling the virtues of this capability when the system was new. Thankfully we had already observed nuclear invaginations on our older Zeiss system, so we knew of their existence. We reverted to the Galvano scanner and the necessary slower scan speeds to reveal these, and then the wider nanocourse network thanks to the improved resolution and signal-to-noise afforded by the Nikon A1R+ system.

5. The purpose of higher temporal image is not only for detection of an event but also to further establish that the event is a bona fide event. Without knowing the kinetics (or several time points on the event, it is not clear what these hotspots are - maybe regions where dye shows greater fluorescence or a movement artefact.

We now fully understand the matter being raised by this Reviewer, and AME apologises for being too dense to pick this up during the first revision. We trust that our new images and plots of the time course of bona fide events, that are tetracaine- and ryanodine-sensitive make the case we had previously failed to address (see revised text, references and figures below).

6. Sparks should have a time course that has a duration at half max of 40-50 ms. The authors indicate that the events are similar to sparks ('As we report, it is MORE than likely that "hotspots" reflect "calcium sparks". The duration, size and amplitude of hotspots are very similar to calcium sparks.') but this cannot be suggested without recording at a minimum of 100 hz.

We now understand the point being made by Reviewer 1. We (AME) had been a little slack in defining the events measured as “Ca²⁺ sparks”. This reference was made all too loosely. We used the phrase to refer to any measure of Ca²⁺ flux arising at ryanodine receptors or clusters thereof, and failed to give due consideration to the fact that the nature of Ca²⁺ sparks had been carefully and quantitatively defined. We have now addressed this

As Reviewer 1 probably suspected, the asynchronous activity, spatiotemporal characteristics and pharmacology of “hotspots” strongly suggests that they reflect low flux and highly variable “quarky calcium release” previously observed at RyR clusters in cardiomyocytes and indicated by trailing events in prolonged “Ca²⁺ sparks” recorded in pulmonary arterial myocytes using line-scan imaging, rather than the briefer (40-50ms), high flux Ca²⁺ sparks previously studied in detail in arterial smooth muscles and other cell types (see revised text, references and figures below).

We have revised the text of the results accordingly, revised Figure 1 and its legend, and provided a new supplementary figure.

Revised text now reads, Page 5 last two lines, through page 6, paragraph 1, lines 1-12

“The fluorescence intensity of nanocourse hotspots exhibited clear stochastic activity (Fig. 1I and Supplementary Fig. 4). Irrespective of nanocourse, at least two discrete levels of hotspot intensity

($\Delta F_H/F_{N0}$; H = hotspot; N0 = nanocourse at 0s) were evident despite the limited temporal resolution at the optimal sampling frequencies used here (0.5Hz; scan speed limited by signal-to-noise). “Gating” times varied in duration, from ~ 2 s to ≥ 10 s for the highest intensity state, with even longer “gating” periods evident for the lower frequency, low intensity sub-state. Transitions ($\Delta F_x/F_0$; hotspot fluorescence at time x and 0s) to the highest intensity state resolved here occurred with a frequency of $\approx 0.16 \pm 0.01$ Hz irrespective of the region of the cell in which they were recorded (80% threshold: peripheral 0.18 ± 0.02 ; extraperinuclear 0.15 ± 0.01 ; perinuclear 0.16 ± 0.02 ; nuclear 0.15 ± 0.02). The asynchronous activity, spatiotemporal characteristics and pharmacology of hotspots suggests that these events most likely reflect low flux and highly variable “quarky calcium release” (QCR; mostly lost due to low signal-to-noise ratios at higher scan speeds⁸) that has previously been proposed to support trailing events that follow prolonged “Ca²⁺ sparks” at RyR clusters in cardiomyocytes⁸, which have also been observed in pulmonary arterial myocytes²⁹, rather than the briefer (40-50ms), higher flux Ca²⁺ sparks previously studied in detail in cardiomyocytes, arterial smooth muscles and other cell types^{3, 30, 31}.”

Figure 1 has been revised accordingly, and now incorporates a new panel (I) which clearly shows “gating” of “hotspots”. The revised section of the legend for Figure 1 now reads, page 27, lines 15-18:

“**I**, Image time series highlights (white rectangle) time-dependent fluctuations in intensity for one single identified hotspot in a different subplasmalemmal nanocourse (indicated by arrow in D, upper panel, right most image), with a record of the time course of “gating” ($\Delta F_H/F_{N0}$; H = hotspot, N₀ = nanocourse at time = 0) from “closed” (C) to “open” (O) states; note also the prolonged sub-state.”

New Supplementary Figure 4 shows further examples of stochastic activity from two different cells. The legend to this figure reads:

“**Supplementary Figure 4 Hotspots within cytoplasmic nanocourses of acutely isolated arterial myocytes exhibit stochastic activity.** **Ai**, Deconvolved confocal z section through the centre of a pulmonary arterial smooth muscle cell loaded with Fluo-4 (pseudocolor applied indicating relative Fluo-4 fluorescent intensity, black = 0, white = maximum). **Aii**, Left hand panel shows image time series for the perinuclear nanocourse identified by the blue ROI shown in (**Ai**), right hand panel shows Fluo-4 fluorescence ratio of the hotspot (F_H) divided by F of the nanocourse (F_{N0}) versus time. **B** as in (**A**) but for a different cell and hotspot within a nuclear invagination.”

Reviewer #2 (Remarks to the Author):

All my critiques have been addressed. In response to several comments, the manuscript has been improved and now give an excellent demonstration of authors’ conclusion.

We thank Reviewer 2 for their kind comments.

REVIEWERS' COMMENTS:

Reviewer #1 (Remarks to the Author):

The authors have taken note of issues that I raised. The responses given do not however address the root of the problem.

In the absence of adequate responses, which would require improved imaging, the conclusions of the MS and the strength of the conclusions drawn require a substantial revision/downscaling.

The main issue that I have relates to the 'Ca events/hotspots' reported in nanocourses.

These supposed events are described as stochastic and of having a specific frequency of occurrence and amplitude.

While the authors recognise that they were slack with their original description of the changes in fluorescence reported as sparks, the change of description of these 'events' to quarky or stochastic does not more adequately describe them.

To describe a change in fluorescence as an event and then to provide features of this event, imaging needs to be performed using a modality that adequately captures this event. Specifically, the sampling time (number of images per second/time between images), should be adequate to capture a baseline, a number of points on the upstroke and downstroke of the event and of course sufficient that the peak of the change in fluorescence is not missed (we would often aim for at least 5). In this MS, sampling frequency is 0.47 Hz. Pixel dwell time may be much shorter but given that images are 2 seconds apart, we cannot determine what is happening in this period. At this capture rate and assuming 5 frames on the rise and fall of the Ca change, then a Ca transient would need to last 20 seconds to be adequately described. Thus, from the experiments described, we cannot conclude any more that that we see a small change in fluorescence at a particular location. The frequency and amplitude can therefore not be deduced. Moreover, we cannot ascribe a gating time to these events. This is absolutely not possible. Whether these events are truly stochastic is also a matter of debate when we cannot determine the duration of the events and without recording for longer and knowing a duration, whether the events have a characteristic frequency or are stochastic cannot be determined. Pharmacological experiments may suggest that the changes in fluorescence at a particular hotspot are 'real' as changes are of 0.05 F/F lower (a very small DF/F) when cells are treated with Tg to deplete stores.

Thus, from an optimistic point of view we can suggest that Ca changes occur and they exist possibly in nanocourses. As we image at a low frequency, we do not know how big these changes (as the peak may be missed) are and how long they last or whether they are simply a non regulated leak.

The author provides a greater description of the galvo imaging. They suggest they have a short pixel dwell time but as images are 2 seconds apart, it is not possible to deduce information re the events/changes in fluorescence.

REBBUTAL TO REVIEWERS' COMMENTS:

Reviewer #1 (Remarks to the Author):

The authors have taken note of issues that I raised. The responses given do not however address the root of the problem.

In the absence of adequate responses, which would require improved imaging, the conclusions of the MS and the strength of the conclusions drawn require a substantial revision/downscaling.

The main issue that I have relates to the 'Ca events/hotspots' reported in nanocourses.

1. *These supposed events are described as stochastic and of having a specific frequency of occurrence and amplitude.*

While the authors recognise that they were slack with their original description of the changes in fluorescence reported as sparks, the change of description of these 'events' to quarky or stochastic does not more adequately describe them.

We understand the Reviewer's concern and have now removed any comment as to the frequency of the fluctuation in Fluo-4 fluorescence at the "hotspots" identified in nanocourses. However, we feel that it is perfectly reasonable to state the amplitude of these fluctuations in Fluo-4 fluorescence because in some cases the transitions from basal to high fluorescence intensity last for 10s or more. There is no doubt that faster events that transition to higher or lower amplitudes will have been missed, but that does not detract from the fact that we do resolve dwell times of sufficient length to confirm the amplitude of the events we can observe.

We also fully understand that, strictly speaking, we cannot assume that the events are truly random and thus stochastic. Therefore, we have removed this adjective from the figures and the text. The revised text now reads as follows (page 7, line 7 to 19):

"Transitions from basal to the highest level of fluorescence intensity ($\Delta F_x/F_0$, mean \pm SEM: peripheral 0.18 \pm 0.02; extraperinuclear 0.15 \pm 0.01; perinuclear 0.16 \pm 0.02; nuclear 0.15 \pm 0.02; n = 7 cells from 7 rats) varied in duration from ~2s to \geq 10s, with even longer dwell times evident for lower frequency, low intensity sub-states. The asynchronous activity, spatial characteristics and pharmacology of hotspots suggests that these events most likely reflect low level, basal Ca²⁺ flux (leak) from the SR via RyRs. However, while RyRs can remain open for many seconds, the fastest gating events are on the millisecond time scale²⁹. Therefore, the development of confocal systems with higher temporal and spatial resolution is required before we can measure the kinetics of hotspots of Ca²⁺ flux characterised here with precision and thus confirm whether they truly represent unitary Ca²⁺ release through RyRs. Nevertheless, it is clear that the Ca²⁺ signalling machinery of subplasmalemmal, extraperinuclear, perinuclear and nuclear nanocourses incorporates unique receptor components, conferring different nanocourses with the capacity to deliver discrete spatially- and functionally-segregated signals."

2. *To describe a change in fluorescence as an event and then to provide features of this event, imaging needs to be performed using a modality that adequately captures this event. Specifically, the sampling time (number of images per second/time between images), should be adequate to capture a baseline, a number of points on the upstroke and downstroke of the event and of course sufficient that the peak of the change in fluorescence is not missed (we would often aim for at least 5). In this MS, sampling frequency is 0.47 Hz. Pixel dwell time may be much shorter but given that images are 2 seconds apart, we cannot determine what is happening in this period. At this capture rate and assuming 5 frames on the rise and fall of the Ca change, then a Ca transient would need to last 20 seconds to be adequately described. Thus, from the experiments described, we cannot conclude any more that that we see a small change in fluorescence at a particular location. The frequency and amplitude can therefore not be deduced. Moreover, we cannot ascribe a gating time to these events. This is absolutely not possible. Whether these events are truly stochastic is also a matter of debate when we cannot determine the duration of the events and without recording for longer and knowing a duration, whether the events have a characteristic frequency or are stochastic cannot be determined.*

We understand the Reviewers' concerns and have revised the text accordingly. See answer to point 1 above for details.

3. *Pharmacological experiments may suggest that the changes in fluorescence at a particular hotspot are 'real' as changes are of 0.05 F/F lower (a very small DF/F) when cells are treated with Tg to deplete stores. Thus, from an optimistic point of view we can suggest that Ca changes occur and they exist possibly in nanocourses. As we image at a low frequency, we do not know how big these changes (as the peak may be missed) are and how long they last or whether they are simply a non regulated leak.*

We understand the Reviewers' point regarding frequency and duration of events, because very brief transitions to higher or lower intensities will be missed at the sampling frequencies used here. However, it is quite clear that we have resolved 2 relatively stable levels of fluorescence that are above baseline. Therefore, we are justified in stating the amplitude of these events. We agree with the reviewer that the events likely represent leak from the SR through RyRs, the gating of which is known to occur at rest; although we would argue that one cannot assume that basal activity is entirely unregulated. See answer to point 1 above for further detailed response.

4. The author provides a greater description of the galvo imaging. They suggest they have a short pixel dwell time but as images are 2 seconds apart, it is not possible to deduce information re the events/changes in fluorescence.

See response to point 1